# Multifunctionality of Nanosized Calcium Apatite Dual-Doped with Li^+^/Eu^3+^ Ions Related to Cell Culture Studies and Cytotoxicity Evaluation In Vitro

**DOI:** 10.3390/biom11091388

**Published:** 2021-09-21

**Authors:** Paulina Sobierajska, Blazej Pozniak, Marta Tikhomirov, Julia Miller, Lucyna Mrowczynska, Agata Piecuch, Justyna Rewak-Soroczynska, Agata Dorotkiewicz-Jach, Zuzanna Drulis-Kawa, Rafal J. Wiglusz

**Affiliations:** 1Institute of Low Temperature and Structure Research, Polish Academy of Sciences, Okolna 2, 50-422 Wroclaw, Poland; j.rewak@intibs.pl; 2Department of Pharmacology and Toxicology, Faculty of Veterinary Medicine, Wroclaw University of Environmental and Life Sciences, C. K. Norwida 31, 50-375 Wroclaw, Poland; blazej.pozniak@upwr.edu.pl (B.P.); marta.tikhomirov@upwr.edu.pl (M.T.); 3Department of Immunology, Pathophysiology and Veterinary Preventive Medicine, Faculty of Veterinary Medicine, Wroclaw University of Environmental and Life Sciences, C. K. Norwida 31, 50-375 Wroclaw, Poland; julia.miller@upwr.edu.pl; 4Department of Cell Biology, Faculty of Biology, Adam Mickiewicz University in Poznan, Uniwersytetu Poznanskiego 6, 61-614 Poznan, Poland; lumro@amu.edu.pl; 5Department of Mycology and Genetics, Wroclaw University, Przybyszewskiego 63, 51-148 Wroclaw, Poland; agata.piecuch@uwr.edu.pl; 6Department of Pathogen Biology and Immunology, Wroclaw University, Przybyszewskiego 63, 51-148 Wroclaw, Poland; agata.dorotkiewicz-jach@uwr.edu.pl (A.D.-J.); zuzanna.drulis-kawa@uwr.edu.pl (Z.D.-K.)

**Keywords:** nanoapatites, Eu^3+^ and Li^+^ ions, rare earth ions, photoluminescence, cytotoxicity, in vitro cell culture studies, protein corona, antibacterial evaluation, theranostics

## Abstract

Li^+^/Eu^3+^ dual-doped calcium apatite analogues were fabricated using a microwave stimulated hydrothermal technique. XRPD, FT-IR, micro-Raman spectroscopy, TEM and SAED measurements indicated that obtained apatites are single-phased, crystallize with a hexagonal structure, have similar morphology and nanometric size as well as show red luminescence. Lithium effectively modifies the local symmetry of optical active sites and, thus, affects the emission efficiency. Moreover, the hydrodynamic size and surface charge of the nanoparticles have been extensively studied. The protein adsorption (lysozyme, LSZ; bovine serum albumin, BSA) on the nanoparticle surface depended on the type of cationic dopant (Li^+^, Eu^3+^) and anionic group (OH^−^, Cl^−^, F^−^) of the apatite matrix. Interaction with LSZ resulted in a positive zeta potential, and the nanoparticles had the lowest hydrodynamic size in this protein medium. The cytotoxicity assessment was carried out on the human osteosarcoma cell line (U2OS), murine macrophages (J774.E), as well as human red blood cells (RBCs). The studied apatites were not cytotoxic to RBCs and J774.E cells; however, at higher concentrations of nanoparticles, cytotoxicity was observed against the U2OS cell line. No antimicrobial activity was detected against Gram-negative bacteria with one exception for *P. aeruginosa* treated with Li^+^-doped fluorapatite.

## 1. Introduction

Synthetic nanoapatites are extensively studied in the area of biomedical applications due to their biocompatibility, high bioactivity and osteoconductive properties in vivo [1]. The biggest advantage of using them as a bioceramic or biomaterial compared to other materials, such as bioglass or glass–ceramic, is their chemical similarity to the inorganic component of the bone tissue. For example, synthetic hydroxyapatite has been applied clinically both as a dense, sintered cement and as a coating on metallic implants [2,3]. Nowadays, researchers are still developing many new areas of using the nanometric hydroxyapatite in orthopedics for drug release and bio-imaging [4,5,6,7]. An ideal diagnostic and therapeutic (theranostics) system should possess the ability to transport active pharmaceutical compounds to target cells or tissues and release them in a controlled manner simultaneously dealing with a real-time monitoring of treatment effect based on photoluminescence properties of rare earth ions (RE^3+^) [8].

In the present research, three analogues of apatite: hydroxyapatite (Ca_10_(PO_4_)_6_(OH)_2_; HAp), chlorapatite (Ca_10_(PO_4_)_6_Cl_2_; ClAp) and fluorapatite (Ca_10_(PO_4_)_6_F_2_; FAp) have been evaluated in terms of cytotoxicity in cell-culture studies. Moreover, these matrices have been modified (while maintaining the crystal structure) using cationic dopants, lithium (Li^+^) and europium (Eu^3+^) ions. Many metal cations can be substituted for calcium ions. Replacement of bivalent Ca^2+^ by other cations, e.g., monovalent (M^+^) or trivalent (M^3+^) ions, cause charge imbalance in the apatite lattice that can be neutralized by creating vacancy defects [8,9]. Whereas, the occurrence of simultaneous substitutions of both M^+^ and M^3+^ cations affects the reduction in vacancy defects, where M^+^ acting as a charge compensation ion [10,11]. Substitution of calcium ions by other cations modifies the cell parameters, which means a contraction or expansion of the lattice constants, depending on ionic radius and charge of the doped ions. The localization of specific ions in the apatitic structure can be generally correlated to their ionic radius. There is a possible substitution of the dopants in both cationic sites: *Ca*_1_ and *Ca*_2_ (see Figure 1), where the first one is smaller in volume than the second one [12]. It leads to the conclusion that a cation with a larger radius than a radius of Ca^2+^ rather should prefer to occupy the *Ca*_2_ site. However, our earlier experiments have shown that indeed for a low concentration range of smaller RE^3+^ ions would preferentially substitute the *Ca*_1_ site, the increase in dopant amount would lead to reverse dependence [10,13]. Probably, the repulsion between atoms in the *Ca*_1_ position would result in the enlargement of the *c*-axis, which is partially inhibited by accommodating metal atoms in the *Ca*_2_ site [14]. Moreover, not only the concentration effect but also the annealing temperature can affect the site preference of the RE^3+^ cations in the apatite lattice. The higher sintering temperature leads to a more preferential incorporation of Eu^3+^ ions at *Ca*_2_ crystallographic position, which is a consequence of the initial migration of Eu^3+^ from the *Ca*_1_ site by a diffusion process [15]. The thermal treatment at above 500 °C activates Ca^2+^ -vacancy-Eu^3+^ migration through the *Ca*_1_ column parallel to the *c*-axis allowing the Eu^3+^ ions (at *Ca*_1_ position) to diffuse through the apatite lattice until obtaining the final distribution approximately equal to 30% at the *Ca*_1_ site and 70% at the *Ca*_2_ site.

In this work, the lithium cation was chosen because it shows (as confirmed in a mice model) a great potential to induce the formation of bone tissue as well as to reduce the risk of fracture in the Li^+^ ions-treated patients [16,17]. Our scientific group has shown that lithium ions can regulate the proliferative activity of mesenchymal stromal cells and their ability to differentiate into various types of cells [18]. From the physicochemical point of view, Li^+^ cation plays an important role as a charge compensator of trivalent lanthanide ion, thus promoting radiative transitions leading to an enhancement in the luminescence intensity [19]. Moreover, the quantum efficiency depends on the type of apatite matrix [20]. Rare earth ions substituted in FAp or ClAp lattice show higher quantum yield in relation to the HAp matrix, where the presence of OH^−^ groups causes strong non-radiative migrations energy. Among of lanthanide ions group, Eu^3+^ cation was selected, because it can be used as a luminescent probe to study the local symmetry of the site(s) that it occupies in the apatite structure and as a high-sensitive, non-invasive tool for luminescence bio-detection in the range of visible light. So far, our scientific group has published research on the influence of Li^+^ ions on the structural and optical properties of FAp as well as Ca-Sr hydroxyapatites activated with Eu^3+^ ions, respectively [10,21].

For biomedical purposes, the toxicity of nanoparticles is a crucial factor in considering their potential in in vivo applications. Since these nanoparticles are expected to interact with living cells, it is important to design them in a way that avoids any adverse effects. The most significant is whether or not modified or unmodified nanoparticles will undergo biodegradation in the cellular environment and what kind of cellular responses will cause such degraded nanoparticles. While in vivo nanoparticles studies require comprehensive and stringent toxicological characterization, an in vitro pathway constitutes the first line of research needed for understanding the primary cytotoxic potential of the particles. This allows a high throughput selection of the most promising materials without unnecessary animal suffering. Typically, in vitro cytotoxicity studies are carried out using cell line models representative of the tissues expected to be the biological targets during in vivo exposure. Such studies are performed under specified conditions (particle concentration and time of exposure) and assess well-defined biological endpoints (e.g., metabolic rate or cell count) [22].

In the current research, the choice of J774.E cell line (murine macrophages) as the in vitro model was based on the fact that under in vivo conditions, macrophages form the primary line of response to particulate matter [23,24]. Thus, they are responsible for the distribution and clearance of nanoparticles and their agglomerates. Moreover, phagocyte interaction with nanoparticulate systems may elicit an inflammatory response that complicates the biocompatibility of some materials [25]. On the other hand, the choice of the human osteosarcoma cell line (U2OS) was due to the fact that it is a cancer cell line derived from bone tissue, which is rich in nanoapatites that play a pivotal role in the extracellular matrix formation. U2OS is an immortal cell line and, therefore, is easier and much more reproducible than primary cells. This cell line belongs to the most widely used models for in vitro studies on the bone tissue [26,27].

Moreover, to expand the evaluation of biocompatibility of the designed biomaterials, the influence of nanoapatites on human red blood cells (RBCs) as well as their interaction with bovine serum albumin (BSA) and lysozyme (LSZ) have been also investigated. Human RBC is the most numerous blood component and a valuable in vitro cell model for screening of biological activity of compounds, especially with the cell membrane-interacting activity [28]. Furthermore, the hemolytic activity of potential blood-contacting compound is an essential parameter in assessing its hemocompatibility [29]. Unfavorable interactions between the newly developed materials and RBCs’ membrane should be carefully analyzed in vitro to prevent any alteration of RBCs’ morphology and function in vivo. Therefore, hemocompatibility is one of the most important criteria that limit the clinical applicability of promising blood-contacting biomaterial. LSZ (cationic) and BSA (anionic) are characterized by opposite charge at pH 7 with isoelectric points 10.70–11.30 and 4.70–5.30, respectively, and are commonly used as model proteins in studies focusing on material–protein interaction. What is more, interactions of LSZ with nanoparticles are extensively studied also because of its antibacterial and possibly also anticancer effect [30,31].

Surgical procedures, for example, the placement of implants, often present certain health risks of bacterial infections through nosocomial strains such as *Pseudomonas aeruginosa*, *Klebsiella pneumoniae* and/or *Escherichia coli* [32,33]. Thus, an implant material should have antimicrobial properties to be protected by bacteria growth and, as a consequence, bacterial infection. It has been noted that the implant materials doped with various metals might exhibit antibacterial activity dependently on the ion concentration. It was shown that hydroxyapatite doped with Eu^3+^ or Li^+^ ions enhanced the bactericidal effect against Gram-positive strains [34,35].

In this paper, the multifunctional properties of nanoapatites structurally modified by Li^+^ and Eu^3+^ ions in comparison with un-doped apatite matrices have been investigated. The obtained materials were comprehensively characterized in terms of their physicochemical and biological properties. The innovation of proposed biomaterials can be associated with the possibility of using them for simultaneous regenerative medicine and in-vivo imaging. Therefore, our research program is dealing with biocompatibility and bioactivity of desired nanocrystalline HAp, ClAp and FAp doped with Li^+^ cation, which influence on cells activity and Eu^3+^ ion used as a bio-imaging probe. Cytotoxicity of obtained materials was evaluated in in vitro cell line models: U2OS, J774.E as well as RBCs to evaluate their hemocompatibility. The interactions of fabricated nanoapatites with BSA and LSZ were also investigated. In the context of the type of dopant, particle size, specific surface area and surface charge, the protein corona evaluation was discussed. Furthermore, the antibacterial properties of studied materials against Gram-negative strains (*E. coli*, *P. aeruginosa* and *K. pneumoniae*) were analyzed.

## 2. Materials and Methods

### 2.1. Preparation of Nanocrystalline Apatites

The nanoapatites: HAp, ClAp and FAp were obtained by a microwave-stimulated hydrothermal method using: Ca(NO_3_)_2_∙4H_2_O (99+% Acros Organics, Geel, Belgium), CaCl_2_∙2H_2_O (99.9% Alfa Aesar, Haverhill, MA, USA), (NH_4_)_2_HPO_4_ (≥98% Avantor Performance Materials Poland S.A, Gliwice, Poland), Eu_2_O_3_ (99.99% Alfa Aesar, Haverhill, MA, USA), Li_2_CO_3_ (99.9% Alfa Aesar, Haverhill, MA, USA) and NH_4_F (98% Alfa Aesar, Haverhill, MA, USA) as starting substrates and NH_3_∙H_2_O (99% was set to 2 mol% Li^+^ and 1 mol% Eu^3+^ in ratio to overall molar content of calcium cations). All reagents used depending on the type of matrix have been gathered in Table 1.

For example, to prepare hydroxyapatite co-doped with Eu^3+^ and Li^+^ (HAp: 1 mol% Eu^3+^, 2 mol% Li^+^), the stoichiometric amount of the Eu_2_O_3_ and Li_2_CO_3_ were digested in an excess of the HNO_3_ (ultrapure Avantor Performance Materials Poland S.A, Gliwice, Poland) to obtain water-soluble europium and lithium nitrates, which, then, were re-crystallized three times in order to remove the excess HNO_3_. Afterwards, an aqueous solution of diammonium hydrogen phosphate was added to the mixture of an aqueous solution of calcium, europium(III) and lithium nitrates. The pH of the suspension was adjusted to 10 by adding ammonia. The reaction condition was maintained at 200 °C for 1.5 h under autogenous pressure of 55 atm in the microwave reactor (ERTEC MV 02-02, Poland). Finally, the resultant precipitate was further dried at 90 °C and then, thermally treated under air atmosphere at 500 °C for 3 h (the heating rate was 3 °C/min).

### 2.2. Materials Characterization

The X-ray powder diffractograms (XRPD) were recorded on a PANalytical X’Pert Pro X-ray diffractometer (Malvern Panalytical Ltd., Royston, UK) equipped with Ni-filtered Cu *Kα*_1_ radiation (*Kα*_1_ = 1.54060 Å, *V* = 40 kV, *I* = 30 mA). The experimental patterns were compared with the Ca_5_(PO_4_)_3_(OH), Ca_5_(PO_4_)_3_Cl and Ca_5_(PO_4_)_3_F standards, respectively, derived from the Inorganic Crystal Structure Database (ICSD). The mean crystallite size was calculated according to the Scherer’s relation:(1)$$D=\frac{0.9\lambda }{coscos\theta \sqrt{\beta^{2}-\beta_{0}^{2}}\;\;}$$
where *D* is the average crystallite size, *λ* denotes the X-ray radiation wavelength, *β* represents a full width at half-maximum of a diffraction line located at *θ*, and *β*_0_ represents a scan aperture of the diffractometer.

The crystallites size was also estimated with the help of Rietveld analysis [36] using an anisotropic approach and Maud 2.68 software [37].

High-resolution transmission electron microscopy (HRTEM) images were performed using a Philips CM-20 SuperTwin microscope (Eindhoven, The Netherlands) with a resolution of 0.25 nm at 200 kV. The sample for HRTEM was prepared by dispersing a small amount of specimen in methanol and putting a droplet of the suspension on a copper microscope gird covered with perforated carbon film. Manual particle size counting (100 measurements) was done by ImageJ software (National Institutes of Health and the Laboratory for Optical and Computational Instrumentation; University of Wisconsin, Madison, WI, USA) based on the known distance (scale bar). A histogram with normal distribution curve has been drawn up using the Origin data analysis software (OriginLab Corporation, Northampton, MA, USA).

The hydrodynamic size of the nanoparticles was determined by the Dynamic Light Scattering (DLS) method with a Zetasizer Nano ZS apparatus from Malvern Instruments operating under an He-Ne 633 nm laser and equipped with the Dispersion Technology Software for data collection and data analysis. The same apparatus was applied for measurements of zeta potential, using a capillary cell, by a combination of electrophoresis and laser doppler velocimetry. The starting concentration of nanoparticles in all prepared suspensions was around 500 µg·mL^−1^. Then, the suspensions were diluted with de-ionized water, 0.05% BSA or 0.05% LSZ water solution until the satisfactory concentration achieved reliable statistics results and excluded errors connected with too high or too low amounts of analyzed objects. Each measurement was repeated three times with fixed concentrations of particles. The hydrodynamic radius (*r_h_*) of the studied samples was determined using the Stokes–Einstein equation:(2)$$r_{h}=\frac{K_{B}T}{6\pi \eta D_{t}}$$
where *K_B_* is Boltzmann’s constant, *T* is temperature, *D_t_* is particle diffusion coefficient, and *η* is solvent viscosity (H_2_O).

The zeta potential (z) of the particles in water suspension was determined based on Henry’s equation:(3)$$z=\frac{U_{E}3\eta }{2\varepsilon f\left(K_{a}\right)}$$
where *UE* is electrophoretic mobility, ε is dielectric constant, *η* is solvent viscosity (H_2_O), and f(*Ka*) is Henry’s function approximately equal to 1.5 (for aqueous suspension).

Overall chemical composition of obtained apatites was performed by Inductively Coupled Plasma Optical-Emission Spectrometry (ICP-OES) using Thermo Scientific iCAP 7000) spectrometer (Waltham, MA, USA). Moreover, the elemental analysis (except too light lithium) was carried out using a Scanning Electron Microscope FEI Nova NanoSEM 230 (Hillsboro, OR, USA) equipped with EDS spectrometer (EDAX PegasusXM4). Up to 10 measurements were made from different random areas to assure satisfactory statistics.

FT-IR spectra were recorded using a Thermo Scientific Nicolet iS50 FT-IR spectrometer (Thermo Waltham, MA, USA) over the wavenumbers 4000–500 cm^−1^ (spectral resolution was set to 4 cm^−1^) in KBr pellets at room temperature, and the background noise was corrected with pure KBr data.

Raman spectra in the range 100–1200 cm^−1^ were carried out using a micro-Raman system Renishaw InVia Raman spectrometer equipped with a confocal DM 2500 Leica optical microscope (Wotton-under-Edge, Gloucestershire, UK), diode laser (λ_exc_ = 830 nm) as an excitation source and thermoelectrically cooled charge-coupled device (CCD) as a detector.

All spectroscopy measurements were performed at room temperature. The emission spectra were recorded using FLS980 Fluorescence Spectrometer from Edinburgh Instruments (Kirkton Campus, UK) equipped with a 450 W Xenon lamp. The emission of a 300 mm focal length monochromator was in the Czerny–Turner configuration. The emission spectra were obtained with ruled grating, 1800 lines per mm blazed at 250 nm. The R928P side window photomultiplier tube from Hamamatsu was used as a detector. A microsecond flashlamp (mF2) was used for the measurements of luminescence decays and a Hamamatsu R928P photomultiplier (Hamamatsu City, Japan) was used as a detector. The effective emission lifetimes (*τ_m_*) were estimated using the equation:(4)$$\tau_{m}=\frac{\int_{0}^{\infty }tI(t)dt}{\int_{0}^{\infty }I(t)dt}≅\frac{\int_{0}^{t^{\max}}tI(t)dt}{\int_{0}^{t^{\max}}I(t)dt}$$
where *I(t)* is the luminescence intensity at time *t* corrected for the background, and the integrals are calculated over the range of 0 < *t* < *t*^max^, where *t*^max^ >> *τ_m_*.

### 2.3. Cytotoxicity Assessment in Osteosarcoma Cell Line and Murine Macrophage

#### 2.3.1. Cell Lines and Culture

Cytotoxicity assessment was carried out on murine macrophage (J774.E) and human osteosarcoma (U2OS) cell lines. Both cell lines were cultured in RPMI-1640 medium (Institute of Immunology and Experimental Therapy, Wroclaw, Poland) supplemented with 10% fetal bovine serum (FBS, Gibco, Amarillo, TX, USA), L-glutamine (Sigma, Welwyn Garden City, UK) and antibiotics (penicillin and streptomycin, Sigma, Hamburg, Germany). Cytotoxicity was assessed by two assays that measure two different endpoints: MTT assay that determines the cellular metabolic activity and trypan blue exclusion assay (TBEA) that is based on counting the actual number of living cells. To exclude possible effects of soluble contaminants or free ionic compounds in the synthesized nanomaterials, the dispersions at the highest nanoparticle concentration were subjected to 2 h centrifugation (at 32,900× *g*), and the supernatants were added to cells as supernatant controls. The lack of adverse effects compared to medium controls indicates that all observed cytotoxicity can be attributed solely to the nanoparticles.

#### 2.3.2. MTT Assay

For the MTT assay, cells were seeded in 96-well plates (TPP, Switzerland) at a density of 10 × 10^3^ (J774.E) or 3 × 10^3^ (U2OS) cells per well and pre-incubated at 37 °C for 24 h in a humidified atmosphere of 5% CO_2_. After that, nanoparticle dispersions were added. Stock dispersions of apatite nanoparticles were prepared based on a simplified version of the NANOGENOTOX dispersion protocol [38]. Nanoparticles were suspended in 0.05% bovine serum albumin (BSA) water solution and bath-sonicated at room temperature for 2 min. Next, the stock solutions were further diluted in 0.05% BSA, and dispersions in complete culture medium were prepared. Cells were exposed to the nanoparticle dispersions for 48 h (5% CO_2_, 37 °C). After that, the tetrazolium salt MTT [3-(4,5-dimethylthiazol-2-yl)-2,5-diphenyl-tetrazoliumbromide] was added. After 2 h of incubation, cells were lysed and left for complete dissolution of purple-colored metabolite. After 24 h, the optical density (OD) was measured using of a microplate reader (Spark 10M, Tecan, Männedorf, Switzerland) at a wavelength of 570 nm (reference 630 nm). The OD of control cells was taken as 100%. The results were obtained from at least 3 independent experiments. To eliminate the possibility of artifacts due to particle interactions with either MTT or formazan, the suspensions were incubated with dispersions (100 µg/mL) in an acellular system. No interactions were observed.

#### 2.3.3. Trypan Blue Exclusion Assay

For the TBEA assay, cells were seeded in 24-well plates (TPP, Switzerland) at a density of 50 × 10^3^ (J774.E) or 20 × 10^3^ (U2OS) cells per well and pre-incubated at 37 °C for 24 h in a humidified atmosphere of 5% CO_2_. After that, nanoparticle dispersions were added (see above). After 48 h of incubation, the culture medium was removed, and cells were washed with phosphate-buffered saline (PBS, Institute of Immunology and Experimental Therapy, Wroclaw, Poland) to remove dead and unattached cells. Next, the remaining cells were detached using 0.25% trypsin in EDTA (Sigma–Aldrich, Taufkirchen, Germany), mixed with 0.4% trypan blue solution (Sigma–Aldrich) and counted in the hemocytometer. In this assay, cells that do not stain blue are considered viable. Cell viability was calculated as a percent of control (treated as 100%). Each assay was made in triplicate, and the results are presented as a mean value (±SD) of three independent assays.

### 2.4. Evaluation of Hemolytic Activity in Human RBCs

#### 2.4.1. Erythrocyte Preparation

Freshly human erythrocytes suspensions (hematocrit/Ht 65%) were purchased from the blood bank in Poznan according to the bilateral agreement No. ZP/907/1002/18. The erythrocytes were washed three times (3000 rpm, 10 min, +4 °C) in 7.4 pH phosphate-buffered saline (PBS—137 mM NaCl, 2.7 mM KCl, 10 mM Na_2_HPO_4_, 1.76 mM KH_2_PO_4_) supplemented with 10 mM glucose. Following washing, RBCs were suspended in PBS buffer at 1.65 × 10^9^ cells/mL (Ht = 15%), stored at 4 °C and used within 5 h.

#### 2.4.2. Hemolysis Assays

Erythrocytes (1.65 × 10^8^ cells/mL, Ht = 1.5%) were incubated in PBS buffer without and with nanoparticles tested (concentrations 0.1 mg/mL and 1 mg/mL) for 60 min at 37 °C in a shaking water bath. Samples with erythrocytes incubated in PBS buffer were taken as the controls. Each sample was repeated three times, and the experiments were repeated three times with RBC from different donors. After incubation, the samples were centrifuged (3000 rpm, 10 min, 4 °C), and released hemoglobin was determined spectrophotometrically at 540 nm (absorption A). The absorption corresponding to a complete hemolysis (absorption B) was acquired after centrifugation of cells treated with ice-cold distilled water. The hemolytic activity of compounds was, then, expressed as a percentage of hemolysis using the following formula:(5)$$\%\;\mathrm{hemolysis}\;=\frac{\mathrm{value}\;\mathrm{of}\;\mathrm{absorption}\;A}{\mathrm{value}\;\mathrm{of}\;\mathrm{absorption}\;B\;}\times 100$$

#### 2.4.3. Erythrocytes Sedimentation Rate under Nanomaterials Treatment

Erythrocytes (1.65 × 10^8^ cells/mL, Ht = 1.5%) were incubated with PBS buffer without and with compounds tested (0.1 mg/mL and 1 mg/mL) in Eppendorf vials for 60 min, at 37 °C. After incubation, the erythrocytes sedimentation rate (ESR) was recorded using a digital camera.

#### 2.4.4. Microscope Studies of Erythrocytes Shape Transformation

The following incubation as above, cells were fixed in 5% paraformaldehyde (PFA) plus 0.01% glutaraldehyde (GA) for 60 min at room temperature (RT). Fixed cells were washed by exchanging of supernatant with PBS buffer, settled on poly-L-lysine-treated (0.1 mg/mL, 10 min, RT) cover glasses and mounted on 80% glycerol. The cover slips were sealed with nail polish. A large number of cells in several separate experimental samples were studied using a Zeiss LSM 510 (AXIOVERT ZOOM) microscope (100 ×/1.4 aperture immersion oil objective, 10 × ocular). Images were acquired using the Zeiss LSM Image Browser program.

### 2.5. Bovine Serum Albumin and Lysozyme Adsorption Apatite Nanoparticles

Bovine serum albumin (BSA) and lysozyme (LSZ) from chicken white egg were obtained from Sigma. Nanoparticles were dispersed in 0.05% BSA or LSZ water solution at three concentrations: 2 mg/mL, 4 mg/mL and 8 mg/mL. After 180 s of bath sonication, the samples were incubated for 4 h at 37 °C under constant agitation. Afterwards, samples were centrifuged (10,000× *g*, 20 °C, 15 min), and the remaining protein concentration was measured in the supernatant using bicinchoninic acid (BCA) protein assay according to the manufacturer’s instruction (ThermoScientific). An additional standard curve for lysozyme was prepared. The results were normalized to blank samples (0.05% BSA or LSZ). The experiment was performed in triplicates.

### 2.6. Antibacterial Evaluation

Antibacterial activity of hydroxyapatites doped with Eu^3+^ and Li^+^ ions was tested using the following strains: *E. coli* ATCC 25922, *E. coli* ATCC 35218, *K. pneumoniae* ATCC 700603, *P. aeruginosa* PAO1 and *P. aeruginosa* ATCC 27853. The strains were cultivated overnight at 37 °C in LB (Luria Broth) medium, centrifuged and suspended in the saline (0.9% NaCl) to obtain optical density of 0.5 McFarland standard. Then, suspensions were diluted 10×, and 10 µL of cell suspension was transferred to 96-well polystyrene plates. The colloidal solutions of hydroxy-, fluor- and chlor-apatites doped with Eu^3+^ (1 mol%), Li^+^ (2 mol%) or Li^+^Eu^3+^ co-doped were diluted in the saline to obtain the final concentration of 100 µg/mL. The solutions (200 μL) were, then, transferred to the wells with bacterial suspensions and incubated for 24 h at 37 °C. The number of survived bacteria was evaluated by colony count (CFU/mL) on Muller–Hinton agar. Bacterial cells incubated with a salt solution were used as a control. Additionally, un-doped apatites were tested for antibacterial activity.

## 3. Results and Discussion

### 3.1. Physicochemical Characterization of Obtained Nanoapatites

According to the XRPD analysis (see Figure 2), the obtained materials are identified as hexagonal apatites belonged to the *P6_3_/m* space group. The positions of the diffraction peaks correspond very well with the positions of the peaks ascribed to the reference standards of the Ca_10_(PO_4_)_6_(OH)_2_ (ICSD-26204), Ca_10_(PO_4_)_6_Cl_2_ (ICSD-24237) and Ca_10_(PO_4_)_6_F_2_ (ICSD-262707). No other peaks were detected, which indicates that the structure of the obtained apatites was stable even after simultaneous doping with ions with different charges (Eu^3+^ and Li^+^). The lack of secondary phases, impurities or amorphous forms confirm the formation of pure phase apatites and high crystallinity of the final products. The diffraction profiles are broadened, which is caused by a small mean crystallites size.

The results of the crystallites size based on Scherer’s relation and Rietveld refinements (see Section 2.2. Materials Characterization) as well as calculated cell parameters were gathered in Table 2. It can be observed that the unit cell parameters decrease when the Eu^3+^ ions are doped. This result is expected, since europium ion (1.12 Å at CN_9_ and 1.01 Å at CN_7_) is smaller than the substituted calcium cations (1.18 Å at CN_9_ and 1.06 Å at CN_7_) [39]. The lithium cation is the smallest among them (0.92 Å at CN_9_). The *a*-axis parameter for all matrices slightly increases with a lithium doping. It can be related to the doping up to 2 mol% Li^+^ and creating cationic vacancies [10]. Above 2 mol% of Li^+^ concentration, a decreasing in the cell volume related to the incorporation of the interstitial Li^+^ into the crystal lattice was observed.

A lack of entrapped water affects the contraction of the *a*-axis observed in the precipitated apatites (water released between 200 °C and 400 °C was irreversibly lost, and the studied materials were sintered at 500 °C) [43]. This was later confirmed on the basis of the FT-IR spectra. In the case of HAp and Fap, lithium causes increasing cell volume. The expected opposite situation was observed in the case of ClAp. Moreover, the reduction in average crystallites size for co-doped apatites was found for both ClAp and FAp. The lack of clear trends suggests that the presence of various anions, i.e., OH^−^, Cl^−^ and F^−^, in the apatite matrix, more precisely in the coordination sphere of calcium crystallographic positions, affects the obtaining cell parameters [12]. The substitution of OH^−^ by F^−^ towards a shift of (2 1 1) and (3 0 0) XRPD reflection to a higher 2θ angle (with the slight deviation in the (1 1 2) position) indicates a contraction of the *a*-axis dimensions (see Figure 3) [44,45]. By contrast, the value of *a*-axis parameter of the ClAp is significantly higher than for HAp (lower-angle shift of (2 1 1) and (3 0 0) planes), unlike in the case of FAp. On the other hand, the *c*-axis parameter increases with fluoride substitution; however, this change is much smaller than for *a*-axis parameter. All the results indicated that contraction and expansion of the *a*-axis parameter of the FAp and ClAp, respectively, are related to ion size mismatch (F^−^—1.32 Å < OH^−^—1.68 Å < Cl^−^—1.81 Å).

The morphology and particle size of the apatites were determined using HRTEM technique. As it can be seen in Figure 4, all studied apatites have similar morphology (elongated nanorods) as well as similar average particle size (HAp: 30 nm × 40 nm, ClAp: 30 nm × 60 nm, FAp: 25 nm × 50 nm). The sample consists of well-crystallized nanograins with a tendency to agglomeration; however, the distribution of the grain size is relatively low. Furthermore, when the selected area electron diffraction (SAED) images reveal the absence of diffused rings and the appearance of intense spots forming a rings, the sample is polynanocrystalline [8]. The rings are present in the images and the spots corresponding to the crystallites (see Figure 4c). The labeled lattice spacing and crystal planes (hkl) (Figure 4b,c and Appendix A) correlate very well with the XRPD data.

The grain size distributions estimated from TEM analysis were compared with those obtained from colloidal suspensions. The results of the hydrodynamic size measurements conducted on water dispersion of particles have been presented in Figure 5 and Table 3. The whole series of nanoapatites (HAp, ClAp, FAp) show a similar distribution of hydrodynamic size in the colloidal suspensions with the maxima at 164 nm for HAp: 1 mol% Eu^3+^, 2 mol% Li^+^, 164 nm in case of ClAp: 1 mol% Eu^3+^, 2 mol% Li^+^ and 154 nm for FAp: 1 mol% Eu^3+^, 2 mol% Li^+^. The difference in the grain size estimation between TEM and DLS is visible. When a dispersed particle moves through a liquid medium a thin, the electric dipole layer adheres to the particle surface, and therefore, this surface influence on intrinsic particle diameter. The thickness of the layer depends on various factors, such as the electrical conductivity of the liquid [46]. Moreover, the use of water on non-stabilizing agent treated particles promote their agglomeration leading directly to the extension of the hydrodynamic particle diameter. Consequently, the choice of dispersing agent is critical for certain biomedical applications. For instance, the way and extent of internalization will be different for single particles and for large agglomerates with an extensive shell called protein corona. The latter scenario usually leads to phagocytosis making macrophages the primary target for the agglomerating particles [47]. Nanomaterials exhibit a high specific surface area, and due to that, nanoparticles in biological fluids strongly adsorb protein on their surface. The composition and size of this protein corona determine how the cell will respond to such a protein interface and, thus, may affect the biodistribution of the nanoparticles and their cytotoxicity. The investigation into this scientific field shows that the formation of the protein corona on nanomaterials is time-dependent exposure. It has been found that corona formation occurs rapidly (<0.5 min) and over time does not transform in composition, but the amount of adsorbed protein changes significantly [48]. Rapid corona formation influence on nanoparticle uptake in the biological system, which ultimately affects the success of the nanomaterial-based therapy. Not only surface properties, but also nanoparticle size, determine the protein corona and both play a significant role in terms of nanosafety issues in a highly dynamic physiological system [49]. In this work, the interaction of studied nanoapatites with model proteins, albumin (BSA) and lysozyme (LSZ) was further discussed (see Section 3.2.3) and correlated with the results presented in Figure 6.

One of the crucial parameters responsible for cell‒nanomaterial interaction is the surface charge of both elements. There is a potential between the surface of the particles and the dispersing liquid, which changes with distance from the surface of the particle. This potential measured for the slipping plane is called zeta potential (*z*) [50]. For clarity, the slipping plane is a boundary within the area of the diffusion layer. When the particle moves, the ions within that boundary move with it, but not any ions located outside that boundary. The measurements were made for two different dispersants, de-ionized water and phosphate-buffered saline (PBS). As it was shown in Table 4, in the case of PBS, the zeta potential reaches higher negative values, which could indicate grater particle stability. Generally, particles with an absolute zeta potential greater than 30 mV are considered stable; however, the stability strongly depends on their size and shape as well as the concentration of the suspension [50]. Particle size and charge are two major factors that play significant roles in the final properties of nanomaterials in biological systems, influencing on release from dosage forms as well as drug circulation and absorption into body membranes [51]. It is known that PBS with pH 7.4 is often used for the assessment of biomaterial in vitro, as its osmolarity and ion concentrations of the solutions are similar to those in the human body. Therefore, when PBS is used in the studies, the zeta potential is affected by not only pH, but also by ionic strength and compositions in the solution. The increase in zeta potential to higher negative values is expected due to an increase in PO_4_^3−^ concentration coming from the buffer. This means that PO_4_^3−^ is a potential-determining ion tightly adhered to the surface, which makes the apatite particles more negatively charged. X. Fan et al. [52] clearly showed the dependence of zeta potential on PO_4_^3−^, Na^+^ and Cl^−^ ions’ concentration (being a component of the PBS buffer). Although Na^+^ and Cl^−^ are not potential-determining ions for calcium phosphates, they also influence on the final zeta potential through adjusting the ion strength in the solution. The authors also studied the use of simulated body fluid (SBF) as an alternative to PBS for the evaluation of biomaterial in vitro, which consequently led them to obtain a positive zeta potential (close to the zero point) of the calcium phosphates.

It is already known that nanoparticles show a high affinity for cellular membranes mainly through electrostatic interactions between them. Cell membranes with a large negatively charged domains should repel negatively charged nanoparticles. However, when there is a possibility of existing cationic sites in the form of clusters, cells can adsorb negatively charged particles [53]. This electrostatic interaction can lead to bending of the membrane favoring endocytosis for cellular uptake [50]. Thus, by modifying the surface charge of the nanoparticles, it is possible to distribute and localize them to specific intracellular targets. For example, nanoparticles with a positive charge are preferentially taken up by usually charged negatively tumors and retained for a longer time interval compared to negatively charged or neutral particles [50]. This also applies to the protein corona, which positively (charged) modifies the surface of nanoparticles. The strategy of using a surface functionalization (changes the surface charge) of the nanoparticles may enhance their interaction with tumor cells, thus making the drug delivery system more effective.

EDS analysis was performed to confirm the elemental composition of the obtained nanoapatites (see Appendix A). The resulting contents of the Eu^3+^, Cl^−^ and F- ions (except Li^+^ ion, which is too light for this measuring technique) to the Ca^2+^ ions were in good agreement with their theoretical molar values. The lithium concentration was determined using the ICP-OES method, and the results are gathered in Table 5. The obtained values confirmed the stoichiometric composition of the synthesized apatites. The estimated molar ratio of cations to phosphorus ions has been very consistent with the theoretical value equal to 1.67.

To obtain more insight into the structure of the obtained materials, the infrared spectra (FT-IR) were measured, and the results are shown in Figure 7. In the IR spectra, the absorption bands characteristic for the hydroxy-, chlor- and fluorapatites are visible. The typical bands of phosphate (PO_4_^3−^) groups were recorded in the region around at 465–478 cm^−1^ (*δ*_2_—the doubly-degenerate bending), 549–620 cm^−1^ (*δ*_4_—the triply degenerate bending), 958–966 cm^−1^ (*ν*_1_—the symmetric non-degenerate stretching vibrations) and 1010–1137 cm^−1^ (*ν*_3_—the asymmetric triply degenerate stretching vibrations) [42,44]. The lack of vibrational transition that appears with the maximum at about 3442 cm^−1^ is associated with combined water loss during the annealing process at 500 °C. The observed stretching vibrational mode at 3572 cm^−1^ and vibrational mode at 633 cm^−1^ indicate the presence of OH^−^ groups in hydroxyapatite structure. In the case of fluorapatite, the existence of active vibrations from OH^−^ groups (belonged to FAp) were not observed, which indicates an ion exchange of OH^−^ for F^−^ during the synthesis of FAp. These OH^−^ modes are also observed for chlorapatite; however, they are much less intense as in the case of HAp, which clearly shows that during the ClAp preparation process, some of OH^−^ ions were not replaced with Cl^−^ ions.

To confirm the formation of pure phase apatite the micro-Raman spectroscopy was applied, and the results are gathered in Figure 7. The spectra of all studied materials contain four characteristic vibrational modes of the phosphate groups [54]. These bands are located at almost the same Raman shift for the same matrix but with different dopants, which indicates that the Li^+^ and Eu^3+^ ions at the selected concentration do not strongly affect the apatite structure. The bands at about 1077, 1047 and 1029 cm^−1^ were assigned to asymmetric *ν*_3_ (P–O) stretching. The most intense peak located at about 964 ± 2 cm^−1^ (depending on the type of matrix) corresponds to the symmetric stretching mode of the phosphate groups *ν*_1_ (PO_4_^3−^). The vibrational bands at about 608, 591 and 580 cm^−1^ are attributed to the *ν*_2_ (PO_4_^3−^) bending modes. The positions of about 447 and 430 cm^−1^ are associated with *ν*_4_ (PO_4_^3−^) bending modes. The overlapping low-intensity bands in the frequency range of 350–100 cm^−1^ confirm the presence of metal–oxygen bonds (*ν*) M–O and (*δ*) O–M–O involved in the structure [10].

The apatites activated with rare earth ions show an intense photoluminescence and, therefore, could be used as a promising material for bio-detection. The red luminescence from the apatites doped and co-doped with Eu^3+^ and Li^+^ ions were presented in Figure 8a. A typical emission spectrum of Eu^3+^ ions located in hexagonal apatite lattice contains five emission bands centered at 575.3 nm (17,382.2 cm^−1^; *^5^D*_0_ → *^7^F*_0_), 591.8 nm (16,897.6 cm^−1^; *^5^D*_0_ → *^7^F*_1_), 612.0 nm (16,339.9 cm^−1^; *^5^D*_0_ → *^7^F*_2_), 653.0 nm (15,313.9 cm^−1^; *^5^D*_0_ → *^7^F*_3_) and 696.9 nm (14,349.3 cm^−1^; *^5^D*_0_ → *^7^F*_4_). The most intensive emission was observed for the hypersensitive *^5^D*_0_ → *^7^F*_2_ transition. The observation of three lines for the *^5^D*_0_ → *^7^F*_0_ transition reveals the presence of non-equivalent crystallographic sites that was previously described at the beginning of this section. It is possible to distinguished them with assigned *C*_3_ (3-fold rotation symmetry) and *C_s_* (reflection symmetry) symmetries using so-called site-selective spectroscopy [55]. Throughout the use of this measuring technique, the preferential ion substitution can be evaluated particularly depending on the type and concentration of dopants. Consequently, it becomes possible to control the luminescence efficiency of novel designed biomaterials based on nanoapatites intended for future theranostic applications. In comparison with traditional using organic dyes, the biomaterials doped with RE^3+^ ions are characterized by narrow absorption/emission lines, long lifetimes of excited states as well as stable and efficient photoluminescence. The organic dyes are usually very fast photo-bleached and chemically degraded as well as their toxicity still being too high for safe medical use.

It is evident from Figure 8 that in the case of matrices with low phonon energies, such as FAp, the emission intensity (a) as well as luminescence lifetime (b) significantly increase. It is also found that upon Li^+^ ions co-doping, the enhanced luminescence of apatite: Eu^3+^ was observed. The substitution with Li^+^ cations strongly affects the crystal field symmetry. The luminescence enhancement induced by Li^+^ ions may be related to the creation of distortions around Eu^3+^ in apatite structure, mainly due to compensation of charge defects and occupation of Li^+^ ions in the interstitial sites (Li^+^ has a considerably smaller radius than Ca^2+^ or Eu^3+^) [19]. The possible charge compensation mechanisms were proposed in our previous paper [10,21]. In addition, it was shown that different concentrations of Li^+^ ions significantly affect the luminescence efficiency of Eu^3+^-doped apatites. The various Eu^3+^ distribution between available cationic sites depending on the concentration of Li^+^ cations was clearly demonstrated based on the Eu^3+^ ions’ luminescence behavior.

### 3.2. Biological Properties

#### 3.2.1. Cytotoxicity Assessment in Osteosarcoma Cell Line

Results of the MTT assay are presented in Figure 9. The left column of graphs shows the metabolic activity of U2OS cells exposed to different concentrations of HAp, ClAp and FAp nanoparticles (from up to down). Neither un-doped nor doped HAp nanoparticles seem to elicit a cytotoxic response in osteosarcoma cells (upper left). On the other hand, both un-doped and doped ClAp nanoparticles seem to elicit some degree of cytotoxicity in osteosarcoma cells, but even at the highest concentration, the viability does not decrease below 50% (Figure 9, middle left). In case of FAp, both Eu^−^ and Li^+^-doped particles visibly decreased cells’ metabolic activity. Surprisingly, the combination of both dopants seems not to cause any cytotoxicity. A high degree of variability precludes drawing firm conclusions, but it is apparent that signs of cytotoxicity appear only at the two highest concentrations. The right column of Figure 9 shows the results of the MTT test for J774.E macrophages. Intriguingly, in all types of apatite nanoparticles, the exposure is associated with an increase in metabolic activity in a concentration-dependent manner. Positive values in MTT assay are typically interpreted as a result of increased cell proliferation. Chen et al. [56] has indeed observed a HAp-induced increase in proliferation of osteoblast cells (MC3T3-E1), but the biologic effects of ClAp or FAp are much less studied. It is known, however, that in some cytotoxicity tests, false positive or false negative results are obtained. This may be due to dye–particle interactions [57] or as a result of changes in numerous enzymes, energy homeostasis or due to oxidative stress [58]. In the present study, cell-independent particle-dye interactions were excluded. To further elucidate whether the results of the MTT test really reflect cell proliferation, we performed the TBEA, an assay where the actual number of living cells is calculated. The results are shown in Figure 10.

A distinct concentration-dependent decrease in viability is seen in U2OS cells (Figure 10, left column). For the un-doped hydroxyapatite, cytotoxicity is observed at higher concentrations (50 µg/mL) compared to all doped nanoparticle types. Anti-proliferative effects of un-doped hydroxyapatite nanoparticles towards different cancer cell lines were described by several authors [59,60]. The mechanism involved is probably associated with particle internalization and subsequent apoptosis [61]. The effect of doping on the cytotoxicity of the nanoapatites used in this study is variable and inconclusive, as no clear trends were identified. In a study by Karthi et al. [62], murine fibroblast cell line L929 tolerated Nd^3+^- and Yb^3+^-doped FAp nanoparticles well (MTT assay). The discrepancy between MTT and TBEA results in the current study is probably associated with the metabolic alterations induced by nanoparticles that may mask and underestimate real cytotoxicity [58]. It is, therefore, important to use more than one test and different endpoints when assessing the biocompatibility of novel materials. In case of J774.E cells, TBEA results suggest no effect on the proliferation even at a very high concentration of 100 µg/mL. Although this suggests high biocompatibility of the nanoparticles under investigation, it should be kept in mind that the results of MTT may still provide important information on metabolic effects of the exposure to nanoparticles. Although not toxic, apatite particles may affect cell metabolism and energy homeostasis in macrophages. Under in vivo conditions, this stimulation might translate into proinflammatory effects or perhaps tissue remodeling.

It is concluded that apatite nanoparticles under investigation are not cytotoxic to J774.E macrophages but may induce slight cytotoxicity in U2OS osteosarcoma cell line at higher concentrations. In macrophages, the investigated nanoparticles may affect (induce) metabolic activity even at non-cytotoxic concentrations, which is probably related to phagocytosis and activation.

#### 3.2.2. In Vitro Hemolytic Activity in Human RBCs

The red blood cells are the most abundant cellular component (99%) of the human circulatory system and are widely used as a cell model in the hemocompatibility studies of different compounds, including nanomaterials, with biomedical applications [63,64,65]. Therefore, new synthesized nanoapatites were evaluated in vitro for their interaction with RBCs’ membrane in the conventional optical microscopy and for their hemolytic activity. Compounds with hemolytic potential prompt the loss of RBCs’ membrane integrity resulting in the release of hemoglobin and in vivo can induce anemia. The results of all experiments clearly show the lack of harmful effect of nanoapatites on the molecular structure and permeability of RBCs’ membrane (i.e., no hemolytic activity—see Figure 11) as well as no effect on RBCs’ discoid shape (see Figure 12 and Table 6) and RBCs’ sedimentation rate (see Table 6 and Figure 13) up to a concentration of 1mg/mL. Whereas, in the case of a higher concentration used, slight aggregates of nanoapatites were detected as attached to the RBCs’ membrane (see Figure 12, arrows). A larger number of aggregates adsorbed on RBCs was observed for the un-doped HAp and ClAp. It should also be noted that RBCs’ incubation with nanoapatites results in their stronger aggregation (compare control RBCs’ images with RBCs’ nanoparticles-treated). The formation of RBCs’ aggregates irregular in size and shape was observed in the presence of many types of nanoparticles, including nanodiamonds [64] or lanthanide-doped core-shell nanomaterials [66]. It was shown that aggregation of RBCs affect their sedimentation pattern [67]. However, apart from RBCs’ aggregating effects observed in the conventional optical microscopy, the nanoapatites studied did not significantly modify the erythrocytes sedimentation rate at both concentrations, with the exception of ClAp (see Figure 13, white circle) at a concentration of 1 mg/mL.

Detrimental interactions of nanomaterials with RBCs, including adsorption of compounds to RBCs’ membrane, RBCs’ aggregation, discocytic RBCs’ shape transformation into spherocytic (swollen, hemolytic cells) and increasing RBCs’ membrane permeability (inducing hemolysis), confirm incompatibility of compounds as blood contacting [66,68]. Our results strongly indicate that nanoapatites does not induce any harmful effects on human RBCs’ membrane structure and function in vitro during both short (60 min) and long (24 h) incubation. Therefore, biocompatibility of apatite nanoparticles studied is confirmed using hemolytic assay as a potent blood-contacting hemocompatible material for biomedical applications. It can be underlined that nanoapatites are attractive to potential medical applications, for example, in drug delivery, bio-imaging or theranostic applications. However, additional studies in vivo should be performed to assess their biocompatibility before considering any medical applications.

#### 3.2.3. Bovine Serum Albumin and Lysozyme Interaction

In the experimental model used, apatites showed much lower affinity towards LSZ (remaining LSZ concentration was mostly around 90% compared to the initial solution) than towards BSA; however, some exceptions were observed (Figure 14). For instance, LSZ was adsorbed to a greater extent on FAp doped with Li^+^ ions and dual-doped with Li^+^ and Eu^3+^ ions and for the latter one. Moreover, LSZ was even undetectable (nd) in the supernatant after the incubation with the highest nanoparticles concentration. A higher affinity toward Li^+^ or Li^+^/Eu^3+^ co-doped nanoparticles is an interesting basis for developing optimal modifications of apatite nanoparticles to provide both antibacterial (LSZ breaks down peptidoglycan of the bacterial cell wall) and bone tissue formation-inducing properties.

An opposite tendency was observed for BSA adsorption—while in general, it adsorbed to a greater extent than LSZ, the lowest remaining protein concentration (thus, higher affinity) was observed for purified apatites or Eu^3+^ doped apatites. The addition of Li^+^ ions decreased protein BSA adsorption in contrary to the effect observed for LSZ. Another noteworthy observation was that ClAp showed the largest difference in the affinity towards LSZ and BSA, adsorbing very low concentrations of LSZ (regardless of Li^+^/Eu^3+^ doping) and the highest concentrations of BSA. Possible causes of this observation include the size of the chloride anion, causing the lower density of the negative charge and, thus, decreasing both the repelling of anionic (BSA) and attraction of cationic (LSZ) protein. Similarly, apatite containing strongly electronegative fluorine anion shows lower a tendency to adsorb BSA compared to ClAp. Observing the tendency that Li^+^ seems to enhance LSZ adsorption in contrast to BSA and the seemingly opposite influence of Eu^3+^ (or, at least no negative influence on BSA adsorption when compared to un-doped apatites), we can see that these interactions cannot be simply explained by the enrichment by cations. However, protein adsorption onto apatite surfaces relies on more factors, such as the inherent surface properties of materials, surface area, surface energy and hydrophobicity but also physiological conditions (including ionic strength) and protein–material interactions such as specific binding at Ca^2+^ and PO_4_^3−^ sites, non-specific binding through hydrogen bonding, etc. [69]. Thus, although as a general rule, positively charged nanoparticles attract proteins with an isoelectric point less than 5 (such as BSA) and negatively charged nanoparticles proteins with a higher isoelectric point [70], some authors observe that the polarity of the protein is not the main factor influencing the binding capability to apatite, e.g., showing similar profiles for BSA and LSZ [69].

Based on detailed research concentrating on the dynamics and kinetics of protein-nanoparticle interaction, other causes of those changes have to be taken into consideration, such as protein concentration, incubation time or even temperature [71]. Another possible cause of observed differences in protein adsorption might be the influence of adsorbed protein on nanoparticle aggregation, which can influence further binding of proteins by reducing the surface of the nanomaterial. This phenomenon was observed by other authors in studies on other types of nanoparticles, e.g., silica [72]. In the cited research, lysozyme was strongly bound by the particles causing their aggregation; however, we cannot exclude the importance of this kind of interaction in our study design, as we did not include solutions with different concentrations of proteins.

Interestingly, the DLS analysis of nanoparticles in protein solutions (see Figure 6) identical to those used for the adsorption test showed an increase in the size (both modal and average) of the nanoparticles dispersed in BSA solution, most probably reflecting the aggregation of nanoparticles and decreased size of the particles in LSZ solution (compared to de-ionized water), probably as a result of a positive effect of the LSZ on nanoparticles stability. In the LSZ adsorption test, LSZ seemed to be bound less effectively to the nanoparticles surface; however, we cannot exclude a potent impact on the stability of the nanoparticles even if small amount of LSZ is bound, which was not detectable in standard protein concentration assays. On the other hand, the design of the protein adsorption assay included centrifugation after the incubation period, so it is also possible that the LSZ binding to nanoparticles is weak and interrupted by further procedures of the test. It is to emphasize that tested nanoparticles showed weaker protein-binding activity, esp. BSA, compared to other particles, as shown in similar tests, for example, modified ferrite particles [73]. Thus, influencing the concentration of nanoparticles used for the test (in lower concentrations, no changes were noted in standard BCA assay). To address the issue in the context of the DLS results, advanced protein corona evaluation methods, for example, electron microscopy, would be necessary.

#### 3.2.4. Antibacterial Evaluation

Since the apatite can enhance bone tissue proliferation, the replacement of bone loss with materials based on hydroxyapatite could facilitate bone regeneration [74]. Moreover, procedures of the implant re/placements often carry a risk of post-surgical infections. Thus, the potential antimicrobial properties of the implanted materials are of special interest.

The antibacterial activity of fluor-, hydroxy- and chlorapatites doped with 2 mol% Li^+^ and 1 mol% Eu^3+^ ions or both dopants were investigated against Gram-negative reference strains, commonly used for drug susceptibility testing or antimicrobial properties of materials (Figure 15).

No antibacterial activity of all tested apatites both doped and co-doped with europium or/and lithium was observed against tested strains, with only one exception for the *P. aeruginosa* ATCC27853 strain treated with fluorapatites doped with 2 mol% Li^+^ or 1 mol% Eu^3+^/2 mol% Li^+^ ions.

As investigated by Iconaru et al. [75], there was no antimicrobial activity of Eu^3+^-doped hydroxyapatites detected against *E. coli* 25922. Our previous study also showed no effect of Eu^3+^-doped hydroxyapatites and chlorapatites against Gram-negative bacteria [76,77]. However, these materials differed in the concentration of Eu^3+^ ions or contained other co-dopants, such as Sr^2+^ ions.

Apatites with lithium dopants have shown efficient osteoblast proliferation, but still, there is not enough data on their antimicrobial properties. Moghanian et al. [34] demonstrated a good antibacterial activity of Li^+^-doped hydroxyapatite bioactive glass against Gram-positive bacteria (MRSA).

## 4. Conclusions

Nanocrystalline apatite analogues un-doped, doped and co-doped with Li^+^ and Eu^3+^ ions were fabricated by using the microwave-assisted hydrothermal method. The XRPD, FT-IR and micro-Raman results have shown that all dopants have been successfully incorporated into the apatite structure causing changes in the lattice parameters. Co-doping with Li^+^ and Eu^3+^ ions caused significant enhancement of luminescence signal that might be useful in future bio-imaging applications. The nano-nature of all studied apatites was confirmed; however, the difference in the grain size was significant. LSZ was found to be a more efficient dispersant in water as compared to BSA. It was found that Li^+^ enhanced LSZ adsorption, whereas Eu^3+^ ions promoted BSA adsorption. FAp doped with both ions showed the highest affinity to LSZ, which may contribute to the antibacterial properties of this material being used as a bone graft substitute.

The biocompatibility of studied apatites was confirmed using U2OS, J774.E lines and human RBCs. The results indicated that nanoapatites do no induce any harmful effects on human RBCs’ membrane structure and function in vitro. A concentration-dependent decrease in cell viability was seen for the U2OS line, and the presence of Eu^3+^ or Li^+^ ions contributed to higher toxicity. The nanoparticle showed no adverse impact on J774.E cell proliferation, even at very high concentrations. Regarding antimicrobial properties, only a slight activity was detected for one *P. aeruginosa* representative treated with Li^+^ -doped fluorapatites, whereas the remaining compounds did not affect selected Gram-negative strains. The presented results suggest that the developed nanoapatite platforms may be a very attractive multimodal nanoagent for theranostic applications covering both regenerative as well as bio-imaging functions. However, extensive in vivo studies should be performed to assess their biocompatibility before considering any medical applications.

## Figures and Tables

**Figure 1 biomolecules-11-01388-f001:**
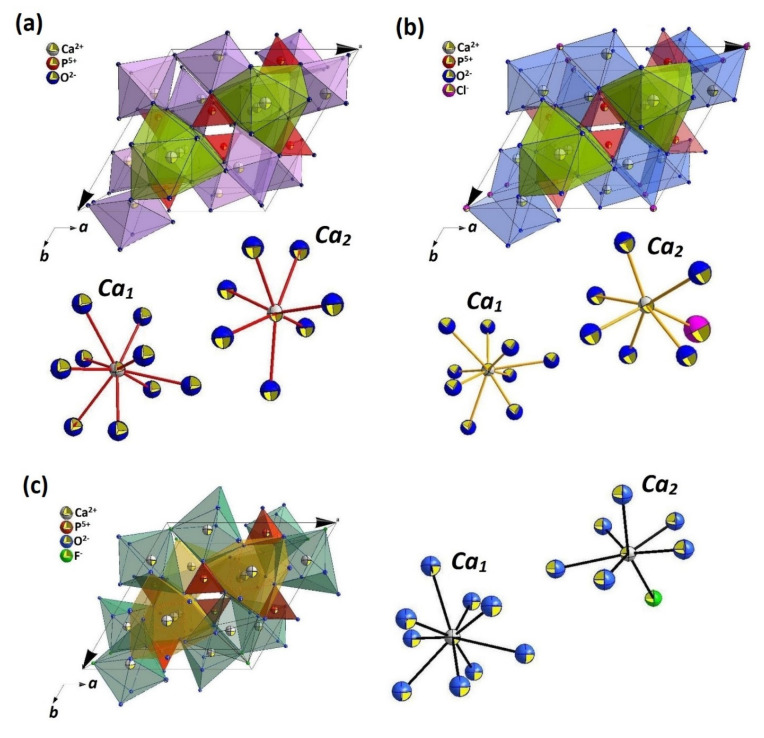
The projection of the Ca_10_(PO_4_)_6_(OH)_2_ (**a**), Ca_10_(PO_4_)_6_Cl_2_ (**b**) and Ca_10_(PO_4_)_6_F_2_ (**c**) unit cells with the indication of the *Ca_1_* and *Ca_2_* coordination spheres.

**Figure 2 biomolecules-11-01388-f002:**
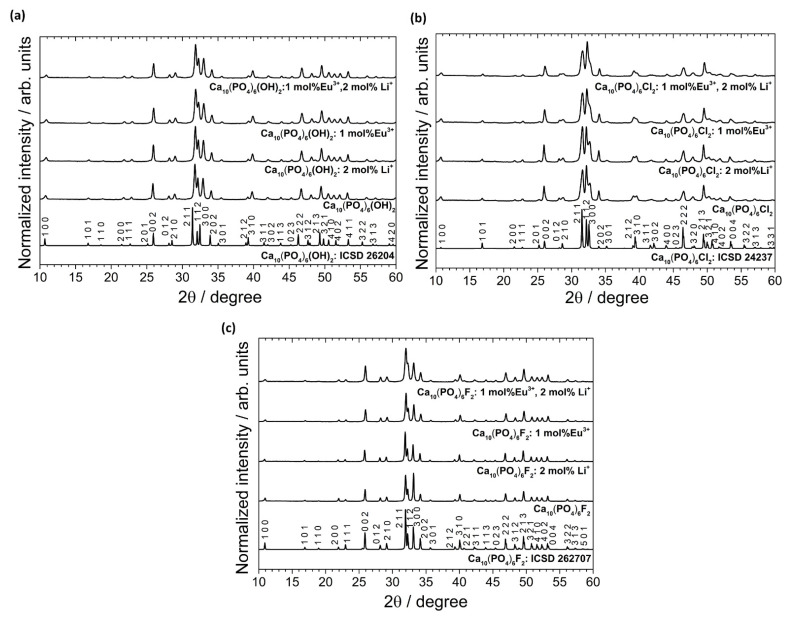
XRPD patterns (with the indication of crystal planes) of the Ca_10_(PO_4_)_6_(OH)_2_ (**a**), Ca_10_(PO_4_)_6_Cl_2_ (**b**), Ca_10_(PO_4_)_6_F_2_ (**c**) un-doped as well as doped and co-doped with 1 mol% Eu^3+^ and 2 mol% Li^+^ ions, prepared at 500 °C.

**Figure 3 biomolecules-11-01388-f003:**
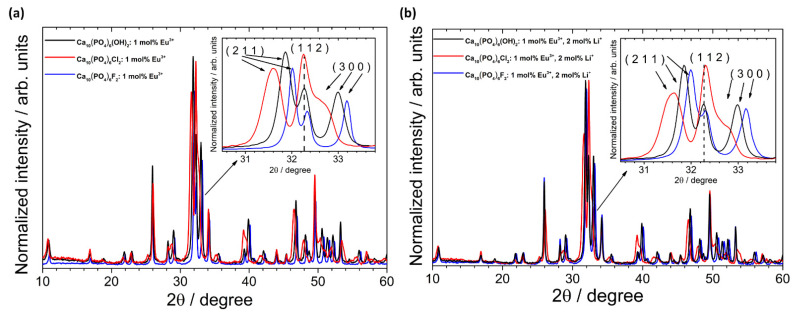
XRPD patterns of the HAp, ClAp and FAp doped with 1 mol% Eu^3+^ (**a**) and co-doped with 1 mol% Eu^3+^ and 2 mol% Li^+^ (**b**) with the indication of lattice planes shift induced by various types of anions (inset).

**Figure 4 biomolecules-11-01388-f004:**
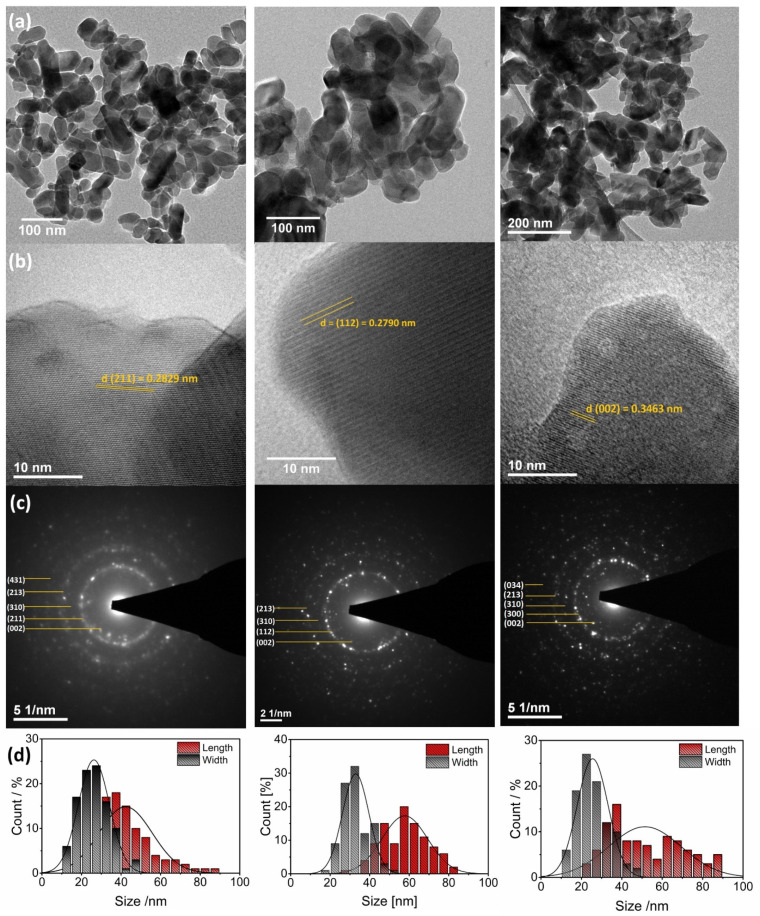
TEM (**a**,**b**) and SAED (**c**) images of the Ca_10_(PO_4_)_6_(OH)_2_ (**left**), Ca_10_(PO_4_)_6_Cl_2_ (**middle**), Ca_10_(PO_4_)_6_F_2_ (**right**) co-doped with 1 mol% Eu^3+^ and 2 mol% Li^+^ prepared at 500 °C with the indication of lattice spacing and crystal planes (hkl). Histogram (**d**) of the grain size distribution (length and width diameters) of the apatites.

**Figure 5 biomolecules-11-01388-f005:**
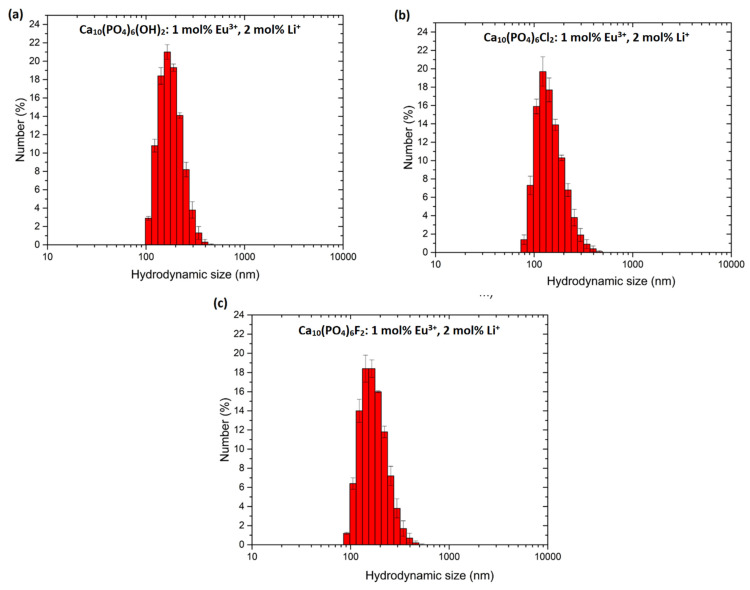
Hydrodynamic size distribution of the nanoapatites water colloidal suspensions: Ca_10_(PO_4_)_6_(OH)_2_ (**a**), Ca_10_(PO_4_)_6_Cl_2_ (**b**) and Ca_10_(PO_4_)_6_F_2_ (**c**) co-doped with 1 mol% Eu^3+^ and 2 mol% Li^+^ ions. Presented values are the mean ± SD of three independent experiments.

**Figure 6 biomolecules-11-01388-f006:**
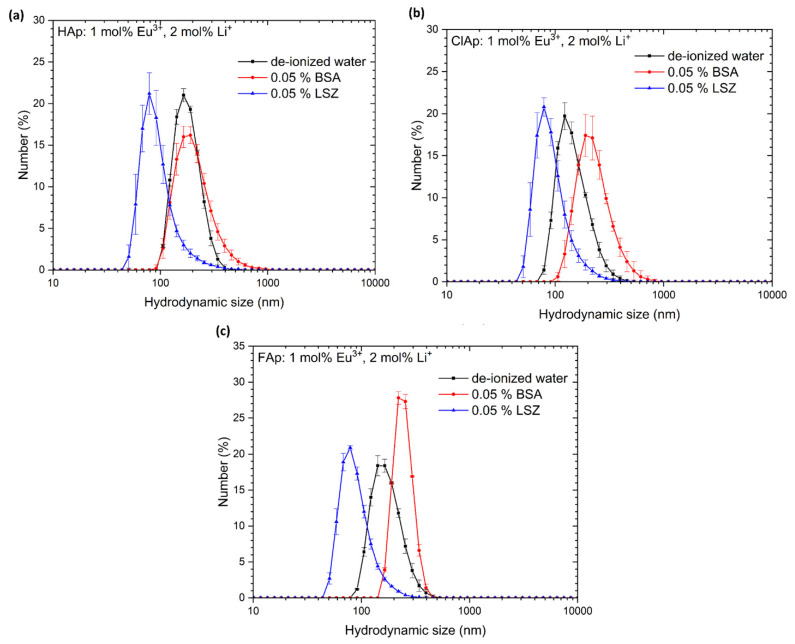
Hydrodynamic size distribution of the nanoapatites (Hap—(**a**), ClAp—(**b**), Fap—(**c**)) co-doped with 1 mol% Eu^3+^ and 2 mol% Li^+^ ions in various colloidal suspensions: de-ionized water, 0.05% BSA and 0.05% LSZ. Presented values are the mean ±SD of three independent experiments.

**Figure 7 biomolecules-11-01388-f007:**
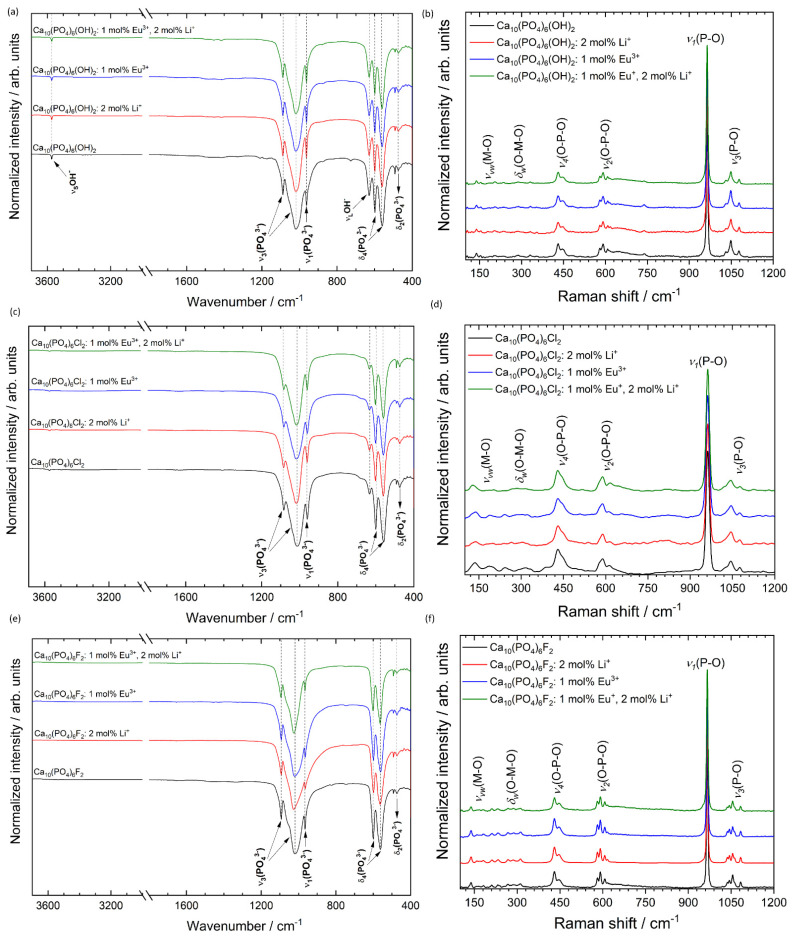
FT-IR (**left**) and micro-Raman (**right**) spectra of the Ca_10_(PO_4_)_6_(OH)_2_ (**a**,**b**), Ca_10_(PO_4_)_6_Cl_2_ (**c**,**d**), Ca_10_(PO_4_)_6_F_2_ (**e**,**f**) co-doped with 1 mol% Eu^3+^ and 2 mol% Li^+^ ions, prepared at 500 °C.

**Figure 8 biomolecules-11-01388-f008:**
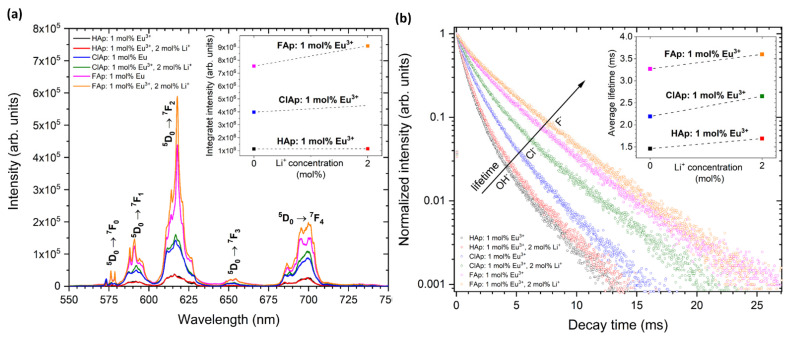
Emission spectra (**a**) of Ca_10_(PO_4_)_6_(OH)_2_, Ca_10_(PO_4_)_6_Cl_2_, Ca_10_(PO_4_)_6_F_2_ doped with 1 mol% Eu^3+^ as well as co-doped with 1 mol% Eu^3+^ and 2 mol% Li^+^ ions under 395 nm excitation with the indication of *^5^D*_0_ → *^7^F_J_* transitions in the range of 550–750 nm, detected at 300 K. The inset shows integral intensity of Eu^3+^ emission as a function of the Li^+^ ions’ concentration. Decay profiles (**b**) of Eu^3+^ ions in different types of apatite matrix, monitored at 618 nm according to the most intense electric dipole transition (*^5^D*_0_ → *^7^F*_2_). The inset shows the average lifetimes of Eu^3+^ ions as a function of the Li^+^ ions’ concentration.

**Figure 9 biomolecules-11-01388-f009:**
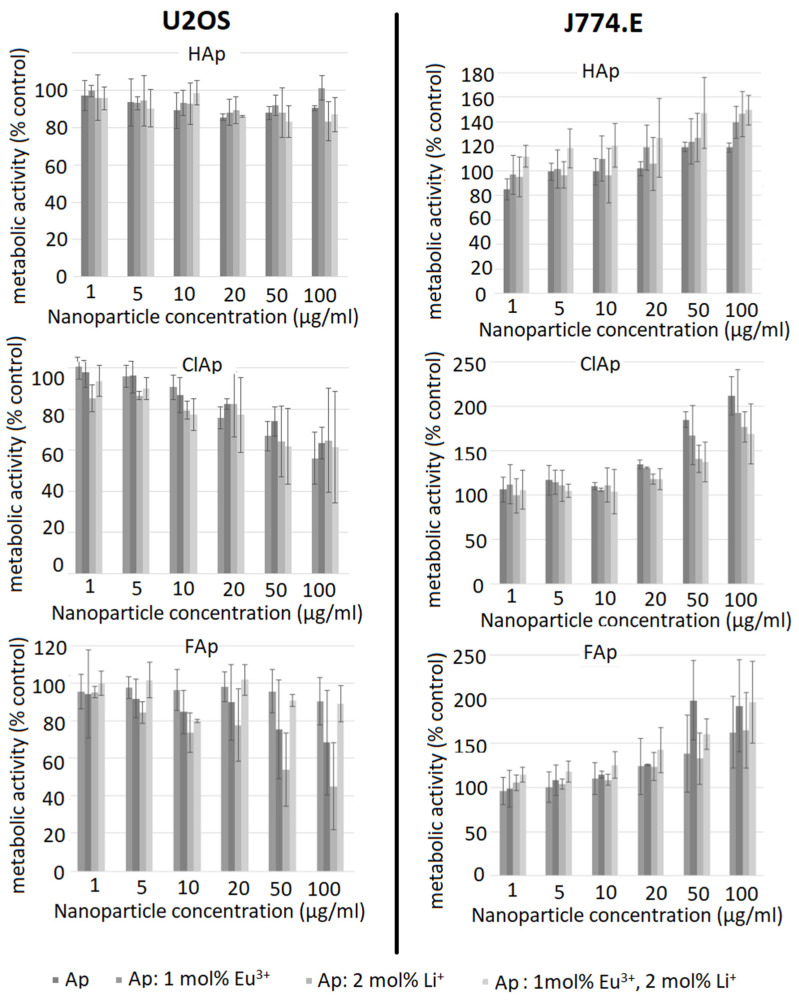
Influence of un-doped nanoapatites (Ap‒ HAp, ClAp, FAp) as well as nanoapatites doped and co-doped with Eu^3+^/Li^+^ ions on metabolic activity of U2OS human osteosarcoma cells (**left panel**) and J774.E murine macrophages (**right panel**), determined by the MTT assay after 48 h exposure.

**Figure 10 biomolecules-11-01388-f010:**
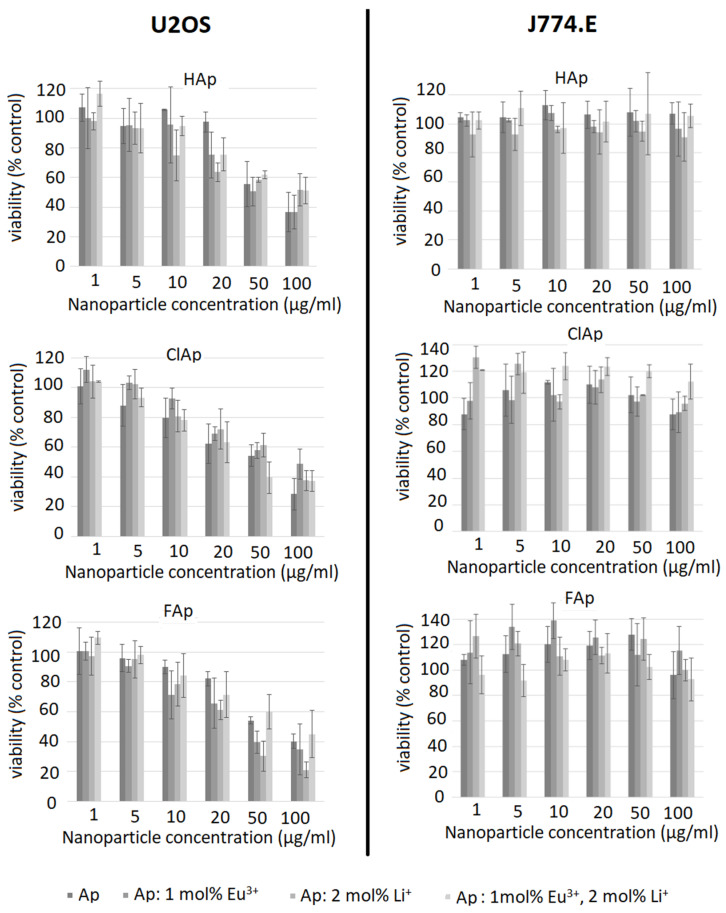
Influence of un-doped nanoapatites (Ap‒HAp, ClAp, FAp) as well as nanoapatites doped and co-doped with Eu^3+^/Li^+^ ions on viability of U2OS human osteosarcoma cells (**left panel**) or J774.E murine macrophages (**right panel**) determined by the trypan blue exclusion assay (TBEA) after 48 h exposure.

**Figure 11 biomolecules-11-01388-f011:**
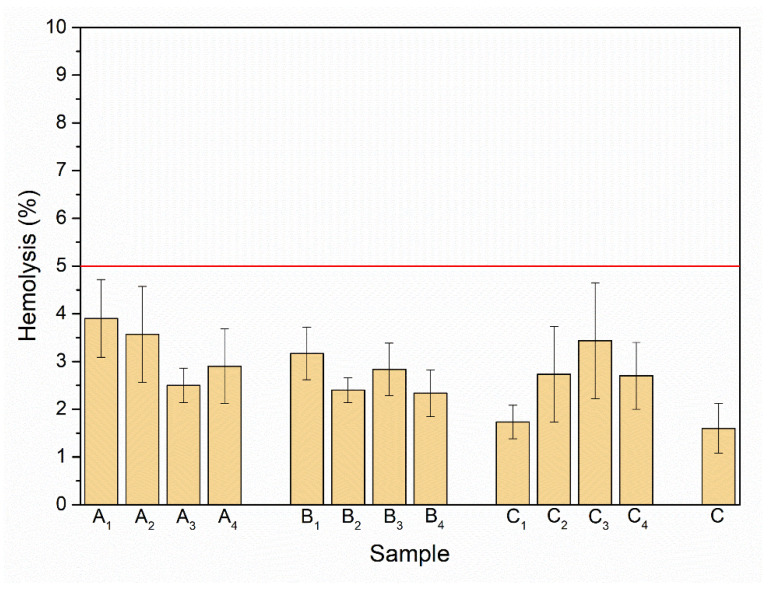
Effect of the nanoapatites (1 mg/mL): HAp (A_1_), ClAp (B_1_) and FAp (C_1_) as well as nanoapatites doped with Li^+^ (A_2_, B_2_, C_2_), doped with Eu^3+^ (A_3_, B_3_, C_3_) and co-doped with Eu^3+^/Li^+^ ions (A_4_, B_4_, C_4_) on selective membrane permeability, i.e., hemolytic activity. Standard incubation conditions (PBS, 60 min, 37 °C; degree of hemolysis lower than 5% corresponding to a lack of hemolytic activity—red line), mean values ± SD, *n* = 9, C—control.

**Figure 12 biomolecules-11-01388-f012:**
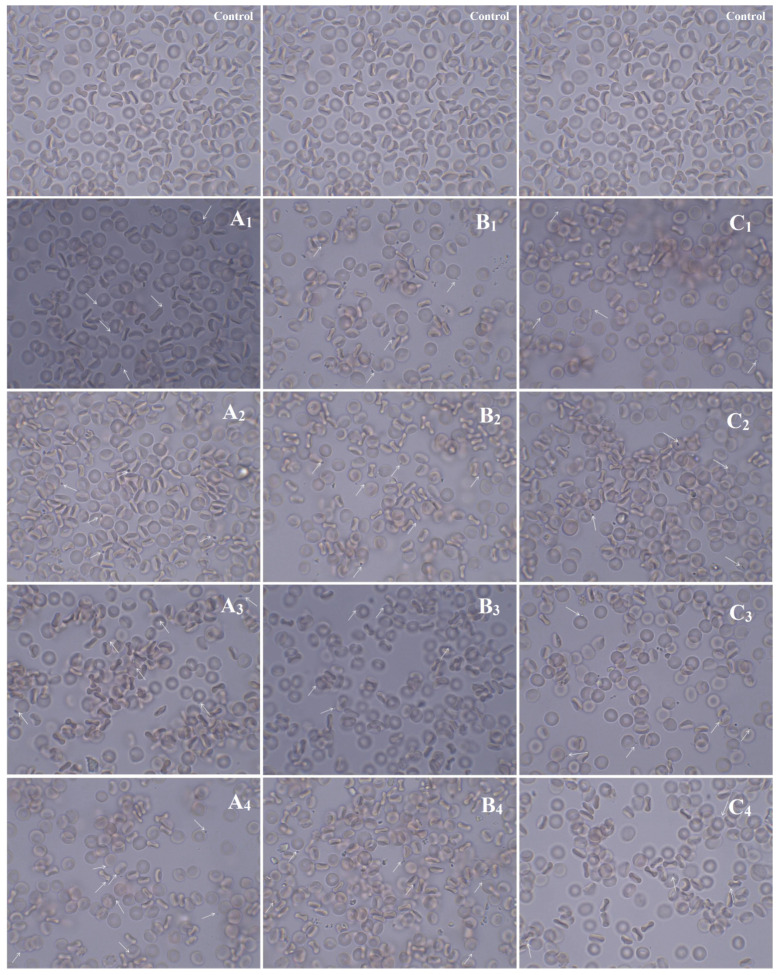
Effect of the nanoapatites (1 mg/mL): HAp (**A_1_**), ClAp (**B_1_**) and FAp (**C_1_**) as well as nanoapatites doped with Li^+^ (**A_2_**,**B_2_**,**C_2_**), doped with Eu^3+^ (**A_3_**,**B_3_**,**C_3_**) and co-doped with Eu^3+^/Li^+^ ions (**A_4_**,**B_4_**,**C_4_**) on the human erythrocyte morphology observed in the light microscopy (60 min, 37 °C) at 100× magnification with immersion oil (10× ocular). Control – cells incubated in PBS buffer. The cells were fixed in a mixture of PFA 5% and GA 0.01%, rinsed 3× in PBS and closed in glycerol. Scale bar in control images is representative for all images presented: 10 µm. Arrows indicate slight aggregates of nanoapatites attached to the RBCs’ membrane.

**Figure 13 biomolecules-11-01388-f013:**
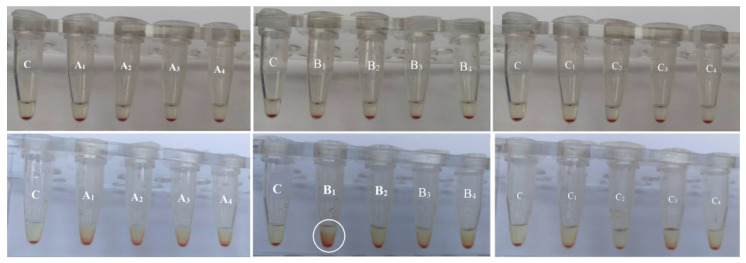
Effect of the nanoapatites: HAp (A_1_), ClAp (B_1_) and FAp (C_1_) as well as nanoapatites doped with Li^+^ (A_2_, B_2_, C_2_), doped with Eu^3+^ (A_3_, B_3_, C_3_) and co-doped with Eu^3+^/Li^+^ (A_4_, B_4_, C_4_) ions on the human erythrocytes sedimentation rate at the concentration of 0.1 mg/mL (**top**) and 1 mg/mL (**bottom**) (60 min, 37 °C). C—PBS buffer, *n* = 3.

**Figure 14 biomolecules-11-01388-f014:**
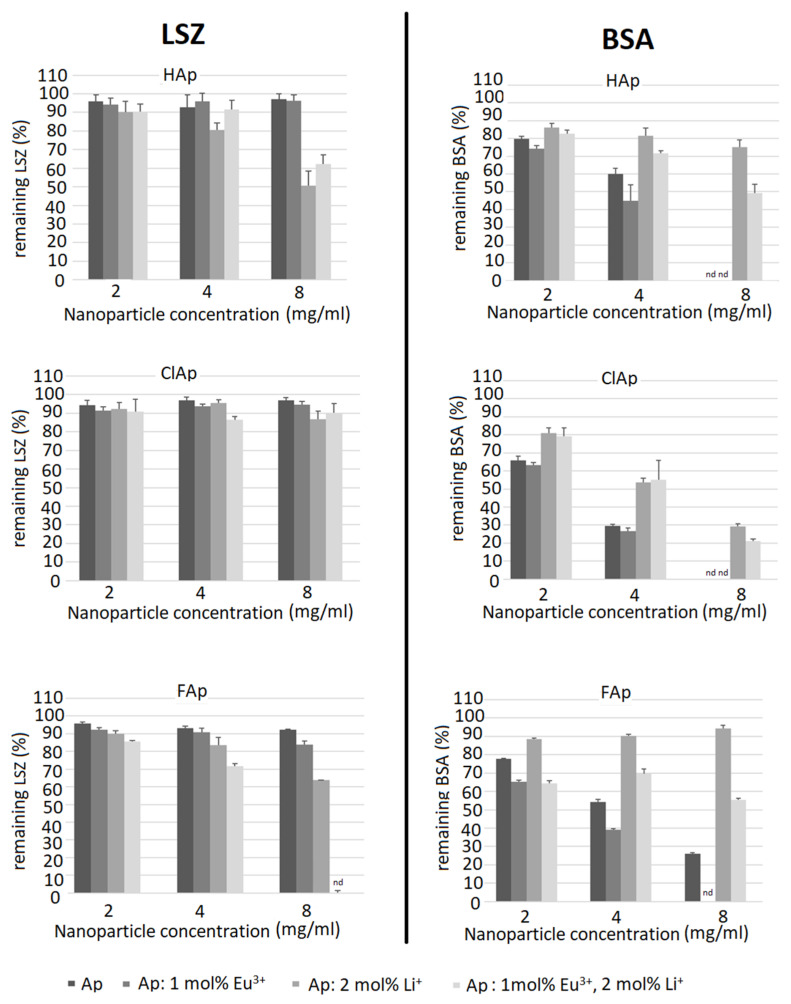
Relative lysozyme (LSZ) (**left panel**) and bovine serum albumin (BSA) (**right panel**) adsorption to un-doped nanoapatites (Ap—HAp, ClAp, FAp) as well as nanoapatites doped and co-doped with Eu^3+^/Li^+^ ions. Nanoparticle dispersions in 0.05% LSZ and BSA were incubated for 4 h, centrifuged and the supernatants were assessed for protein concentration. Solutions of 0.05% LSZ and BSA were used as control (100% protein concentration).

**Figure 15 biomolecules-11-01388-f015:**
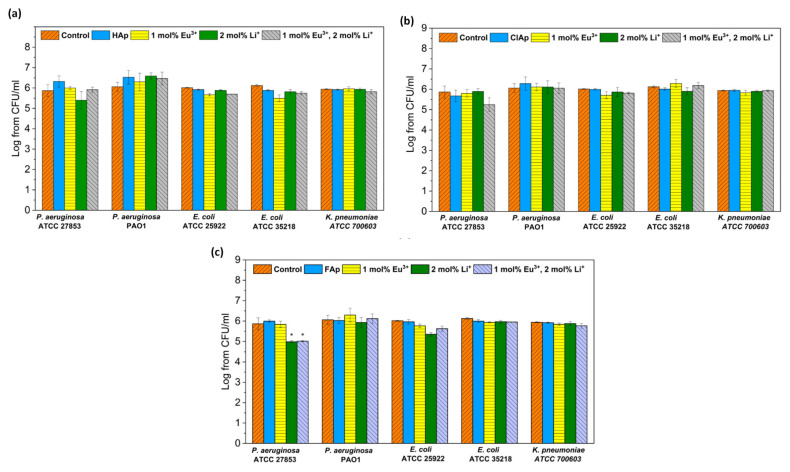
Antibacterial effect of hydroxyapatites (**a**), chlorapatites (**b**) and fluorapatites (**c**) doped with Eu^3+^ and Li^+^ ions (concentration of 100 μg/mL) against Gram-negative strains: *P. aeruginosa*, *E. coli* and *K. pneumoniae*; mean ± SD, *n* = 3; *—statistically different from the control (*p* < 0.05).

**Table 1 biomolecules-11-01388-t001:** Reagents used for the synthesis of nanoapatites (+ means reagent used, − means reagent not used).

	Ca(NO_3_)_2_·4H_2_O	CaCl_2_·2H_2_O	(NH_4_)_2_HPO_4_	NH_4_F	Eu_2_O_3_	Li_2_CO_3_
HAp	+	−	+	−	+	+
ClAp	−	+	+	−	+	+
FAp	+	−	+	+	+	+

**Table 2 biomolecules-11-01388-t002:** Unit cell parameters (a,c), cell volume (V), mean crystallites size calculated according to the Scherer (S) and Rietveld (R) method as well as refine factor (R_W_) for the Ca_5_(PO_4_)_3_Z (where Z—OH^−^, Cl^−^, F^−^) doped and co-doped with 1 mol% Eu^3+^ and 2 mol% Li^+^ powder prepared at 500 °C.

Sample	a (Å)	c (Å)	V (Å^3^)	Size R (nm)	Size S (nm)	R_w_ (%)
Ca_10_(PO_4_)_6_(OH)_2_, single crystal [40]	9.424 (4)	6.879 (4)	529.09 (4)	–	–	–
Ca_10_(PO_4_)_6_(OH)_2_: 1 mol% Eu^3+^	9.400 (3)	6.870 (7)	525.79 (2)	42.02	47.80	1.1
Ca_10_(PO_4_)_6_(OH)_2_: 1 mol% Eu^3+^,2 mol% Li^+^	9.403 (8)	6.870 (0)	526.13 (3)	56.13	55.17	1.5
Ca_10_(PO_4_)_6_Cl_2_, single crystal [41]	9.52 (3)	6.85 (3)	537.64 (3)	–	–	–
Ca_10_(PO_4_)_6_Cl_2_: 1 mol% Eu^3+^	9.51 (0)	6.84 (4)	536.94 (8)	34.38	34.98	1.9
Ca_10_(PO_4_)_6_Cl_2_: 1 mol% Eu^3+^, 2 mol% Li^+^	9.51 (8)	6.83 (0)	535.84 (9)	28.58	25.57	1.8
Ca_10_(PO_4_)_6_F_2_, single crystal [42]	9.3672 (1)	6.8844 (1)	523.15 (1)	–	–	–
Ca_10_(PO_4_)_6_F_2_: 1 mol% Eu^3+^	9.3481 (8)	6.8705 (9)	519.97 (0)	86.39	75.52	2.5
Ca_10_(PO_4_)_6_F_2_: 1 mol% Eu^3+^, 2 mol% Li^+^	9.3557 (6)	6.8723 (1)	520.94 (5)	50.35	57.49	2.1

**Table 3 biomolecules-11-01388-t003:** Main hydrodynamic size of nanoapatites colloidal suspensions: Ca_10_(PO_4_)_6_(OH)_2_, Ca_10_(PO_4_)_6_Cl_2_ and Ca_10_(PO_4_)_6_F_2_ co-doped with 1 mol% Eu^3+^ and 2 mol% Li^+^ ions, measured in de-ionized water, 0.05% BSA and 0.05% LSZ based on DLS technique.

Sample	Typical Hydrodynamic Size * (nm)	Average Hydrodynamic Size (nm)	PdI **
**de-ionized water**			
Ca_10_(PO_4_)_6_(OH)_2_: 1 mol% Eu^3+^, 2 mol% Li^+^	164 ± 1	251 ± 11	0.28 ± 0.02
Ca_10_(PO_4_)_6_Cl_2_: 1 mol% Eu^3+^, 2 mol% Li^+^	164 ± 2	202 ± 12	0.18 ± 0.03
Ca_10_(PO_4_)_6_F_2_: 1 mol% Eu^3+^, 2 mol% Li^+^	153 ± 2	237 ± 8	0.25 ± 0.03
**0.05% BSA**			
Ca_10_(PO_4_)_6_(OH)_2_: 1 mol% Eu^3+^, 2 mol% Li^+^	177 ± 2	451 ± 17	0.41 ± 0.01
Ca_10_(PO_4_)_6_Cl_2_: 1 mol% Eu^3+^, 2 mol% Li^+^	205 ± 5	284 ± 5	0.46 ± 0.08
Ca_10_(PO_4_)_6_F_2_: 1 mol% Eu^3+^, 2 mol% Li^+^	235 ± 2	307 ± 16	0.28 ± 0.02
**0.05% LSZ**			
Ca_10_(PO_4_)_6_(OH)_2_: 1 mol% Eu^3+^, 2 mol% Li^+^	78 ± 2	225 ± 4	0.45 ± 0.06
Ca_10_(PO_4_)_6_Cl_2_: 1 mol% Eu^3+^, 2 mol% Li^+^	80 ± 1	192 ± 17	0.35 ± 0.02
Ca_10_(PO_4_)_6_F_2_: 1 mol% Eu^3+^, 2 mol% Li^+^	79 ± 0.2	161 ± 3	0.33 ± 0.03

* Size corresponding to the maximum of the monodisperse domain (mode of the size distribution). ** Polydispersity index (giving a measure of dispersion in size).

**Table 4 biomolecules-11-01388-t004:** Zeta potentials of nanoapatites series in de-ionized water, PBS, BSA and LSZ suspensions.

Sample	Zeta Potential (mV) De-Ionized Water	Zeta Potential (mV) PBS	Zeta Potential (mV) BSA	Zeta Potential (mV) LSZ
Ca_10_(PO_4_)_6_(OH)_2_	−16.5 ± 0.5	−19.7 ± 0.6	−18.5 ± 0.4	16.7 ± 0.2
Ca_10_(PO_4_)_6_(OH)_2_: 2 mol% Li^+^	−11.0 ± 0.9	−18.9 ± 0.6	−22.4 ± 0.6	22.9 ± 0.2
Ca_10_(PO_4_)_6_(OH)_2_: 1 mol% Eu^3+^	−9.4 ± 0.2	−17.1 ± 0.9	−17.5 ± 0.6	16.7 ± 0.3
Ca_10_(PO_4_)_6_(OH)_2_:1 mol% Eu^3+^, 2 mol% Li^+^	−10.9 ± 0.1	−18.9 ± 0.3	−18.9 ± 0.7	17.8 ± 0.6
Ca_10_(PO_4_)_6_Cl_2_	−8.7 ± 0.3	−17.6 ± 0.5	−21.6 ± 0.8	25.6 ± 0.9
Ca_10_(PO_4_)_6_Cl_2_: 2 mol% Li^+^	−7.3 ± 0.1	−20.6 ± 0.2	−21.1 ± 0.9	21.6 ± 0.8
Ca_10_(PO_4_)_6_Cl_2_: 1 mol% Eu^3+^	−5.8 ± 0.1	−21.3 ± 0.5	−20.0 ± 0.5	21.9 ± 0.9
Ca_10_(PO_4_)_6_Cl_2_: 1 mol% Eu^3+^, 2 mol% Li^+^	−8.4 ± 0.1	−20.2 ± 0.3	−26.9 ± 0.7	22.8 ± 0.2
Ca_10_(PO_4_)_6_F_2_	−3.6 ± 0.1	−22.0 ± 0.7	−16.3 ± 0.8	26.1 ± 0.8
Ca_10_(PO_4_)_6_F_2_: 2 mol% Li^+^	−15.4 ± 0.2	−24.1 ± 0.8	−19.9 ± 0.7	28.8 ± 0.1
Ca_10_(PO_4_)_6_F_2_: 1 mol% Eu^3+^	−10.2 ± 0.3	−20.0 ± 0.2	−17.7 ± 0.5	29.5 ± 0.5
Ca_10_(PO_4_)_6_F_2_: 1 mol% Eu^3+^, 2 mol% Li^+^	−10.2 ± 0.3	−18.7 ± 0.1	−16.3 ± 0.8	28.9 ± 0.3

**Table 5 biomolecules-11-01388-t005:** The element content in the obtained hydroxyapatite measured by ICP-OES technique.

Scheme	Li (wt.%)	Eu (wt%)	Ca (wt%)	P (wt%)	$\frac{n_{Li}}{n_{\left(Li+Ca+Eu\right)}}$ (mol%)	$\frac{n_{Eu}}{n_{\left(Li+Ca+Eu\right)}}$ (mol%)	$\frac{n_{\left(Li+Ca+Eu\right)}}{n_{P}}$
Ca_10_(PO_4_)_6_(OH)_2_: 1 mol% Eu^3+^, 2 mol% Li	0.138 ± 0.01	1.51 ± 0.1	39 ± 2	18 ± 1	2.02 ± 0.04	0.94 ± 0.02	1.65 ± 0.02
Ca_10_(PO_4_)_6_Cl_2_: 1 mol% Eu^3+^, 2 mol% Li	0.133 ± 0.01	1.45 ± 0.1	37 ± 2	18 ± 1	1.83 ± 0.04	0.92 ± 0.02	1.66 ± 0.02
Ca_10_(PO_4_)_6_F_2_: 1 mol% Eu^3+^, 2 mol% Li^+^	0.137 ± 0.01	1.50 ± 0.1	38 ± 2	18 ± 1	1.83 ± 0.04	0.97 ± 0.02	1.66 ± 0.02

**Table 6 biomolecules-11-01388-t006:** The effect of the nanoapatites: HAp (A_1_), ClAp (B_1_) and FAp (C_1_) as well as nanoapatites doped with Li^+^ (A_2_, B_2_, C_2_), doped with Eu^3+^ (A_3_, B_3_, C_3_) and co-doped with Eu^3+^/Li^+^ (A_4_, B_4_, C_4_) ions on human erythrocytes (60 min, 37 °C) at the concentration of 1 mg/mL; at lower concentrations (0.1 mg/mL), no effect of the compounds on RBCs was detected.

The Effect of Nanoapatites on Human Erythrocytes *				
Sample	Sample Code	Dominated Erythrocytes Shape	Binding to RBCs’ Membrane	Effect on ESR
Ca_10_(PO_4_)_6_(OH)_2_	A_1_	D	++	+
Ca_10_(PO_4_)_6_(OH)_2_: 2 mol% Li^+^	A_2_	D	+	+
Ca_10_(PO_4_)_6_(OH)_2_: 1 mol% Eu^3+^	A_3_	D	+	+
Ca_10_(PO_4_)_6_(OH)_2_:1 mol% Eu^3+^, 2 mol% Li^+^	A_4_	D	+	+
Ca_10_(PO_4_)_6_Cl_2_	B_1_	D	++	++
Ca_10_(PO_4_)_6_Cl_2_: 2 mol% Li^+^	B_2_	D	+	−
Ca_10_(PO_4_)_6_Cl_2_: 1 mol% Eu^3+^	B_3_	D	+	−
Ca_10_(PO_4_)_6_Cl_2_: 1 mol% Eu^3+^, 2 mol% Li^+^	B_4_	D	+	−
Ca_10_(PO_4_)_6_F_2_	C_1_	D	+	−
Ca_10_(PO_4_)_6_F_2_: 2 mol% Li^+^	C_2_	D	+	−
Ca_10_(PO_4_)_6_F_2_: 1 mol% Eu^3+^	C_3_	D	+	+
Ca_10_(PO_4_)_6_F_2_: 1 mol% Eu^3+^, 2 mol% Li^+^	C_4_	D	+	−
PBS buffer (Control)	C	D	no	−

* Presented values are the mean ± SD of three independent experiments with erythrocytes from different donors; abbreviations: D—discocytes (normal cells, such as control cells). (+)—weak effect observed, (++)—strong effect observed, (‒)—no effect observed; ESR—erythrocyte sedimentation rate.

## Data Availability

Data are available from the authors upon request.

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
