# Peer review of "Multifunctionality of Nanosized Calcium Apatite Dual-Doped with Li+/Eu3+ Ions Related to Cell Culture Studies and Cytotoxicity Evaluation In Vitro"

_biomolecules, 2021, doi:10.3390/biom11091388_

Round 1

Reviewer 1 Report

see the attachment

Reviewer 2 Report

The sugestions to improve the manuscript are shown in the attached file. In the case of some figures like Fig. 2. XRD powder patterns, it will be useful the indexing of the main peaks; in the caption of Fig.12, it is important to say something about the arrows shown in the images. In References section, please use the same style of citation of the journals: acronyms or not acronyms. 

Reviewer 3 Report

Comment 1
Lines 37 - 38
The authors must delete a line

Comment 2
Lines 880 - 881
Something missing (journal: Nanoscale?) 

Comment 3
Lines 892 - 893
Something missing (journal: Journal of Solid State Chemistry?) 

Comment 4
Line 73
would lead to reverse dependence [10] [13]. Probably,
replace
would lead to reverse dependence [10,13]. Probably,

Comment 5
Line 98
respectively [10] [21].
replace
respectively [10,21].

Comment 6
Line 918
Something missing

Comment 7
Line 133
hemocompatibility[29].
replace
hemocompatibility [29].

Comment 8
Lines 129 - 130
on human red blood cells (RBCs) as well as their interaction
with bovine serum albumin (BSA) and lysozyme (LSZ),

replace

on human Red Blood Cells (RBCs) as well as their interaction 
with Bovine Serum Albumin (BSA) and lysozyme (LSZ),

Comment 9
Line 165
Line 185

The authors must renumber the subsections.

3.1. Preparation of nanocrystalline apatites
3.2. Materials characterization 
replace
2.1. Preparation of nanocrystalline apatites
2.2. Materials characterization 

Comment 10
The numbering of the equations should be at the end of the line on the right side of the text.

Comment 11
Line 200
lution of  0.25nm at 200 kV.
The authors must delete a space between the of and the 0.25nm 

Comment 12
Lines 255 - 256
3.3. Cytotoxicity assessment in osteosarcoma cell line and murine macrophage 
3.3.1. Cell lines and culture 

replace
2.3. Cytotoxicity assessment in osteosarcoma cell line and murine macrophage 
2.3.1. Cell lines and culture 

Also, It's not so good to start the subsection at the bottom of the page without using text

Comment 13
Line 269
Line 287
Line 299
Line 300
Line 307
Line 319
Line 324
3.3.2. MTT assay
3.3.3. Trypan blue exclusion assay
3.4. Evaluation of hemolytic activity in human RBCs
3.4.1. Erythrocyte preparation
3.4.2. Hemolysis assays
3.4.3. Erythrocytes sedimentation rate under nanomaterials treatment
3.4.4. Microscope studies of erythrocytes shape transformation
replace
2.3.2. MTT assay
2.3.3. Trypan blue exclusion assay
2.4. Evaluation of hemolytic activity in human RBCs
2.4.1. Erythrocyte preparation
2.4.2. Hemolysis assays
2.4.3. Erythrocytes sedimentation rate under nanomaterials treatment
2.4.4. Microscope studies of erythrocytes shape transformation

Comment 14
The authors must check the numbering of the section and subsection

Line 299 3.4. Evaluation of hemolytic activity in human RBCs

and 

Line 333 3.4. Bovine serum albumin and lysozyme adsorption apatite nanoparticles

Comment 15
Line 369
The authors must check if (see Subsection 3.2. Materials characterization) is right

Comment 16
Line 381
The authors must check if  (see Figure 7). is right

Comment 17
According to the journal's instructions 
All Figures, Schemes and Tables should be inserted into the main text close to their first citation 
and must be numbered following their number of appearance 
(Figure 1, Scheme I, Figure 2, Scheme II, Table 1, etc.).

Comment 18
Major problem
Line 400 Table 2
Line 504 Table 2

Two Tables 2!!!

Comment 19
Line 982
Biomaterial
replace
Biomaterials

Comment 20
Line 465
H. Fan et al.[52] clearly
replace
X. Zhu et al. [52] clearly

Comment 21
Line 988
Something missing (journal?) 

Comment 22
Line 524
are shown in Figure 7. In
Why underlined?

Comment 23
Line 529
stretching vibrations) [41] [45].
replace
stretching vibrations) [41,45].

Comment 24
Line 592
our previous paper [10] [21].
replace
our previous paper [10,21].

Comment 25
Lines 596 - 597
2.2. Biological properties 
2.2.1. Cytotoxicity assessment in osteosarcoma cell line

Comment 26
Line 637
thors [59][60]. The 
replace
thors [59,60]. The 

Comment 27
Line 657
Line 719
Line 787
2.2.2. In vitro hemolytic activity in human RBCs
2.2.3. Bovine serum albumin and lysozyme interaction 
2.2.4. Antibacterial evaluation
The authors must change the numbering.

Comment 28
Lines 805 - 806 
against Gram-negative bacteria [76]
[77]. However,
replace
against Gram-negative bacteria [76,
77]. However,

Comment 29
Lines 843
plications.[40] 
replace
plications. 

Comment 30
Increase the number of the reference papers including (primarily) from biomolecules.
The authors use 0 paper from biomolecules journal / 1 paper from MDPI Journals / 77 papers from journals (References)
Τhe number for papers from MDPI journals
is considered insufficient (in reviewer's opinion).

Reviewer 4 Report

Ref: biomolecules-1336809, entitled "Multifunctionality of nanosized calcium apatite dual-doped with Li+/Eu3+ ions related to cell culture studies and cytotoxicity evaluation in vitro".

In this work, the authors investigated the multifunctional properties of nanoparticles structurally modified by Li+ and Eu3+ ions in comparison with un-doped apatite matrices. The obtained materials were comprehensively characterized in terms of their physicochemical and biological properties. The content of this work is enrich and interesting. However, before reaching the quality and robustness standards of Biomolecules, it would require minor improvements, especially as regards the following general aspects:

  1. In the “3.1. Preparation of nanocrystalline apatites” section, more experimental details should be added.

(1) How long was the materials maintained at 200 ℃ ?

(2) Heating rate and atmosphere in the thermally treatment process ?

  1. To better observe the XRPD spectrums (Figure 2) of the as-prepared samples, the stack diagram is recommended. Moreover, the XRPD standard peaks of Ca10(PO4)6(OH)2 (ICSD-26204), Ca10(PO4)6Cl2 (ICSD-24237) and Ca10(PO4)6F2 (ICSD-262707) should be labeled in the Figure 2.

  1. The lattice spacing and crystal planes should be measured and labeled in the HRTEM pictures. The crystal planes corresponding to the diffraction ring should be also labeled in the SAED pictures.

  1. What method is used to calculate the grain size distributions estimated from TEM analysis? This point should be mentioned in the manuscript.

  1. To better convince the proposed conclusions for readers, the following points should be supported by relative literature.

(1) Page 9, line 375-376: It can be related to the doping up to 2 mol% Li+ and creating cationic vacancies.

(2) Page 18, line 544-549: affect the apatite structure. The bands at about 1077, 1047, and 1029 cm-1 were assigned to asymmetric ν3(P-O) stretching. The most intense peak located at about 964 ± 2 cm-1 (depending on the type of matrix) corresponds to the symmetric stretching mode of the phosphate groups ν1(PO43-). The vibrational bands at about 608, 591 and 580 cm-1 are attributed to the ν2(PO43-) bending modes. The positions of about 447 and 430 cm-1 are associated with ν4(PO43-) bending modes.

Round 2

Reviewer 3 Report

The paper deals with Multifunctionality of nanosized calcium apatite dual-doped 2 with Li+/Eu3+ ions related to cell culture studies and cytotoxicity 3 evaluation in vitro. 
According to the reviewer, the paper is worth publishing at biomolecules Journal, 
but some corrections are needed and then the paper can be accepted for publication in the journal.
While the authors have made considerable research effort, 
the presentation of the paper and the results must be proved. 
Additionally make the following corrections to the manuscript:

Comment 1
Line 149
The authors must rephrase. It is not very good to use the word "we".

Comment 2
Line 252
The authors must move the number of the equation 4 in the middle of the equation.

Comment 3
Line 379: Reference [39]

then Table 2: Reference [43]

and then Line 393: Reference [40]

According to the journal's instructions:
References must be numbered in order of appearance in the text (including citations in tables and legends) 
and listed individually at the end of the manuscript. 

The authors must renumber.

Comment 4
Line 421
(Figure 4 b,c and Table S1)
replace
(Figure 4 b,c and Table S1, supplementary materials)

Comment 5
Line 474
H. Fan et al. [52] clearly
replace
X. Zhu et al. [52] clearly

Comment 6
Line 597
low 50% (Figure 9, middle left). ). In case
replace
low 50% (Figure 9, middle left). In case

Comment 7
First the authors mention Table 3 in the text and then Figure 13, 
while the order presented in the paper is first Figure 13 and then Table 3.
Move the Table 3 before Figure 13.  

Comment 8
Line 800
It's not so good to start the sub-section at the bottom of the page without using text

Comment 9
Lines 824 - 826
The authors must give more details also for Table S1
